# Neoadjuvant anti-OX40 (MEDI6469) therapy in patients with head and neck squamous cell carcinoma activates and expands antigen-specific tumor-infiltrating T cells

Rebekka Duhen [1✉], Carmen Ballesteros-Merino[1], Alexandra K. Frye [1], Eric Tran [1],
Venkatesh Rajamanickam [1], Shu-Ching Chang[2], Yoshinobu Koguchi[1], Carlo B. Bifulco[1,3], Brady Bernard[1],
Rom S. Leidner [1], Brendan D. Curti[1], Bernard A. Fox[1], Walter J. Urba[1], R. Bryan Bell [1] &
Andrew D. Weinberg [1,4✉]

Despite the success of checkpoint blockade in some cancer patients, there is an unmet need to improve outcomes. Targeting alternative pathways, such as costimulatory molecules (e.g. OX40, GITR, and 4-1BB), can enhance T cell immunity in tumor-bearing hosts. Here we describe the results from a phase Ib clinical trial (NCT02274155) in which 17 patients with locally advanced head and neck squamous cell carcinoma (HNSCC) received a murine anti-human OX40 agonist antibody (MEDI6469) prior to definitive surgical resection. The primary endpoint was to determine safety and feasibility of the anti-OX40 neoadjuvant treatment. The secondary objective was to assess the effect of anti-OX40 on lymphocyte subsets in the tumor and blood. Neoadjuvant anti-OX40 was well tolerated and did not delay surgery, thus meeting the primary endpoint. Peripheral blood phenotyping data show increases in CD4+ and CD8+ T cell proliferation two weeks after anti-OX40 administration. Comparison of tumor biopsies before and after treatment reveals an increase of activated, conventional CD4+ tumor-infiltrating lymphocytes (TIL) in most patients and higher clonality by TCRβ sequencing. Analyses of CD8+ TIL show increases in tumor-antigen reactive, proliferating CD103+ CD39+ cells in 25% of patients with evaluable tumor tissue (N = 4/16), all of whom remain disease-free. These data provide evidence that anti-OX40 prior to surgery is safe and can increase activation and proliferation of CD4+ and CD8+ T cells in blood and tumor. Our work suggests that increases in the tumor-reactive CD103+ CD39+ CD8+ TIL could serve as a potential biomarker of anti-OX40 clinical activity.

[1] Earle A. Chiles Research Institute, Providence Cancer Institute, Portland, OR, USA. [2] Medical Data Research Center, Providence Saint Joseph's Health, Portland, OR, USA. [3] Molecular Genomics Laboratory, Providence St. Joseph Health, Portland, OR, USA. [4] AgonOx, Inc., Portland, OR, USA. ✉email: rebekka.duhen@providence.org; andrew.weinberg@providence.org

Head and neck squamous cell carcinoma (HNSCC) is the seventh most common cancer worldwide and is characterized by a high rate of therapeutic resistance[1–3]. Recurrence rates are ~50% in locally advanced HNSCC[4], despite aggressive treatment involving surgery, radiation, and chemotherapy. Checkpoint inhibitor (CI) antibodies targeting programmed cell death protein 1 (PD-1) or programmed death ligand-1 (PD-L1) have improved overall survival for patients with metastatic HNSCC, and have been FDA approved in conjunction with chemotherapy as first-line treatment. However, only 10–20% of patients benefit from PD-1/PD-L1 blockade[5–8].

Neoadjuvant immunotherapy improved survival compared to adjuvant treatment in preclinical models[9] and a number of investigators are currently testing CI's in the neoadjuvant setting for the treatment of HNSCC (NCT02296684, Ravindra Uppaluri; NCT02764593, Robert Ferris). Feasibility and safety of checkpoint blockade prior to surgery have been demonstrated clinically in urothelial carcinoma of the bladder[10], resectable non-small cell lung cancer (NSCLC) independent of PD-L1 expression[11], stage III/IV melanoma[12,13], and glioblastoma multiforme (GBM)[14–16]. Interestingly, in melanoma, neoadjuvant treatment with anti-PD-1 results in major pathologic responses and brisk TIL infiltration that correlates with disease-free survival[17].

The role of immune costimulatory agonist agents in the neoadjuvant setting in cancer patients has not been previously investigated. This is the first report of a human T-cell agonist antibody specific for OX40 administered in the neoadjuvant setting. OX40 (CD134), a member of the tumor necrosis factor receptor superfamily, is part of a potent costimulatory pathway that can enhance T-cell memory, proliferation, survival, and antitumor activity[18–21]. In contrast to other costimulatory molecules, such as CD27 and CD28, OX40 is absent on resting naïve and memory T cells, and induced only after T-cell receptor (TCR) engagement[22]. Immune profiling studies comparing TIL and peripheral blood from patients with HNSCC demonstrated expression of OX40, as well as PD-1 and CTLA-4, on T cells isolated from the tumor compared to blood, suggesting that targeting these pathways independently or in combination may result in therapeutic advantages[23]. A first in human phase I clinical trial with this antibody to OX40 (MEDI6469) demonstrated acceptable toxicity, increased proliferation of both CD4+ and CD8+ T cells, and led to regression of at least one metastatic lesion in 12 of 30 patients treated[24]. Our laboratory has also demonstrated that anti-OX40 treatment reduces recurrences when given prior to surgery in murine models[25]. Altogether, these results suggested that delivering anti-OX40 prior to surgery in HNSCC patients may confer clinical benefit. Furthermore, this presents an ideal opportunity to analyze the immune effects of anti-OX40 therapy in the tumor.

Here, we show that administering an anti-OX40 antibody at various intervals prior to definitive surgical resection in 17 patients with stage II-IVA HNSCC (Clinical Trial ID#NCT02274155) is safe. The optimal immunologic activation occurs 2 weeks after anti-OX40 administration in the tumor and periphery. Immune monitoring using flow cytometry, quantitative multiplex immunochemistry (mIHC), and TCR sequencing to interrogate the effects of anti-OX40 in the tumor, suggests that increases in the tumor-reactive CD8+ TIL population may serve as a biomarker of clinical activity.

## Results

**Neoadjuvant anti-OX40 treatment in patients with HNSCC is safe.** Between December 2014 and April 2017, 17 of 19 patients who were consented and assessed for eligibility were enrolled in the trial and received anti-OX40 at 0.4 mg/kg on days 1, 3, and 5

(Fig. 1a and Supplementary Data 1 for the study protocol). Three patients were enrolled into cohort 1 (day 8 surgery), nine patients were in cohort 2 (day 12 surgery), four patients were in cohort 3 (day 19 surgery), and one patient was enrolled at day 26, a time point that was removed from the protocol subsequently (Supplementary Table 1a). The average age was 60 years (SD ± 8.9 years) and clinical staging based on the American Joint Committee on Cancer (AJCC) 7th edition ranged from stage II to IVA disease; tumors involved the oral cavity ($N = 6$), oropharynx ($N = 9$), hypopharynx ($N = 1$), or larynx ($N = 1$)[26] (Supplementary Table 1a, b). Tumor HPV status, assessed by p16 IHC testing, was diffusely positive in six of nine oropharyngeal tumors and the remainders were negative. All patients underwent definitive surgical resection, neck dissection, appropriate reconstruction, and pathologic risk-adapted adjuvant radiation or chemoradiation per standard of care. Three patients had prior radiation therapy related to their cancer. All other patients were treatment naive (Supplementary Table 1b). In 16 of 17 patients, a tumor biopsy specimen was obtained prior to treatment, and at the time of definitive surgery, the primary tumor was collected as well as metastatic and draining lymph node(s) in some patients. PBMC were collected before and after treatment in all patients.

Anti-OX40 was well-tolerated and did not delay or cause toxicity prior to surgery. There were no grade III or IV adverse events (AE) due to anti-OX40 (Supplementary Table 2). Grade III or IV surgical complications, as assessed by the Clavien–Dindo classification, occurred in 29% ($N = 5$) of patients, which included a non-ST elevated myocardial infarction (managed with stent placement), sepsis, and cellulitis (managed successfully with antibiotics), and acute respiratory failure (managed uneventfully), none of which were attributed to anti-OX40 administration (Supplementary Table 3). One patient developed meningitis postoperatively, which resolved with antibiotics and steroids. In general, the spectra of toxicities observed in this trial were similar to what was reported in the first in human trial, together with lymphopenia in most patients[24]. Complete blood counts (CBC) revealed transient lymphopenia along with higher neutrophil counts in all patients, potentially due to migration of lymphocyte subsets into lymphoid organs and tissues followed by a rebound (Fig. 1b and Supplementary Fig. 1a). Lymphocyte counts showed the greatest decrease (by fold change) in D12 patients, and neutrophil counts were highest in patients that underwent surgery at D19. Taken together, anti-OX40 administration was safe and well-tolerated in all patients enrolled in the trial, with transient lymphopenia 2 weeks following antibody infusion.

**T-cell proliferation in peripheral blood following anti-OX40 administration peaks between D12 and D19.** Immune activation was investigated by assessing phenotypic changes in PBMC at different timepoints following anti-OX40 administration (baseline, day of surgery, day 34, and day 55; Supplementary Fig. 2a, b). The change in percentage of conventional CD4+ (Tconv), CD8+, and Foxp3+ (Treg) cells in patients was analyzed over time, but no significant changes were observed in these subsets in the D8, D12, or D19 cohorts. The percentage of Treg cells increased slightly after anti-OX40 in both the D12 and D19 cohorts (Fig. 1c and Supplementary Fig. 1b), albeit not significantly ($P = 0.0585$, D12; $P = 0.0525$, D19). Using Ki-67 as a proliferation marker, we detected an increase in proliferation between baseline and D12 and D19 in CD4+ Tconv cells ($P = 0.0074$, D12; $P = 0.0015$, D19) as was observed in the initial phase I clinical trial using MEDI6469[24] (Fig. 1d and Supplementary Fig. 1c, f). We did not observe a correlation between baseline and day of surgery (DOS) OX40 expression by Tconv CD4+ cells and immune activation/proliferation after anti-OX40 administration

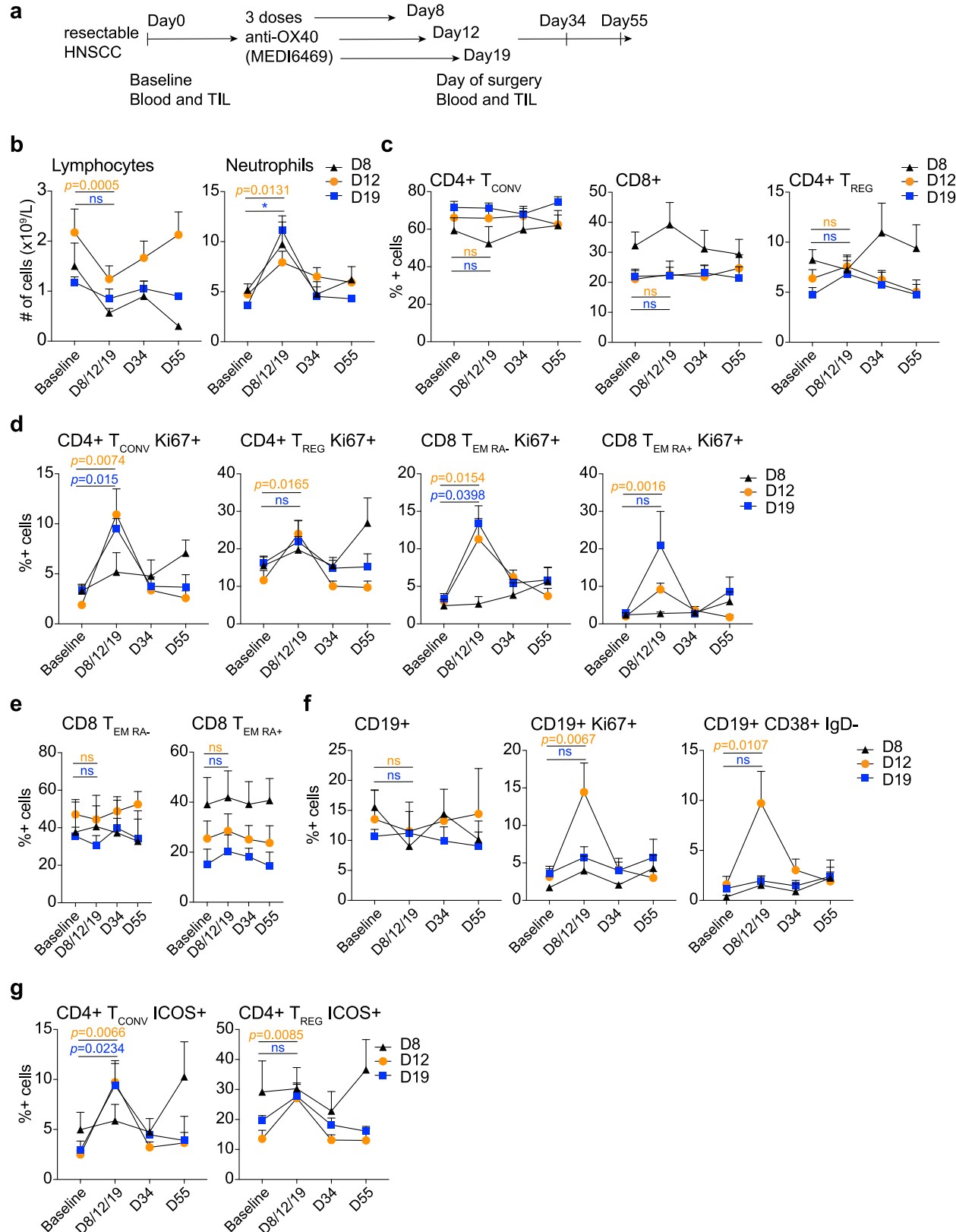

(Supplementary Fig. 1d), and OX40 expression was unchanged during the course of the 2-month monitoring (Supplementary Fig. 1e). Proliferation also increased in Foxp3+ Treg cells, albeit at a lower level (2.6-fold) compared to conventional CD4+ T cells (Tconv), which increased 6.8-fold in the D12 cohort. Proliferation of CD8+ TEMRA− and effector memory CD45RA+ (TEMRA+)

T cells consistently peaked with a four- to sixfold increase between D12 and D19 (Fig. 1d and Supplementary Fig. 1f) ($P = 0.0154$ TEMRA−, $P = 0.0016$ TEMRA+, D12; $P = 0.0398$ TEMRA−, $P = 0.12$ TEMRA+, D19), while the frequencies of each of these subsets remained constant during the time monitored (Fig. 1e and Supplementary Fig. 1g). The percentage of B

**Fig. 1 Immune activation after OX40 administration in head and neck squamous cell carcinoma (HNSCC).** Patients with HNSCC cancer were given three doses of neoadjuvant anti-OX40 (MEDI6469) at 0.4 mg/kg, followed by resection at days 8, 12, or 19. **a** Trial schema of administration of anti-OX40 in the neoadjuvant setting in patients with HNSCC. **b** Blood lymphocyte and neutrophil counts were assessed at baseline, day of surgery (D8, D12, or D19), D34, and D55. The mean + SEM of the absolute cell counts per liter are shown for each subset. **c–e** Change in percentages of viable, CD3+ T cells. The gating strategy is outlined in Supplementary Fig. 2a, b. **c** Average percentages of CD4+ Tconv, CD8+, and CD4+ Treg cells during and after anti-OX40 administration measured by flow cytometry. **d** Summary of the average percentage of Ki-67 expression during OX40 treatment in CD4+ Tconv memory cells, CD4+ Treg cells, CD8+ $T_{EM\ RA-}$, and $T_{EM\ RA+}$ cells. **e** Average percentages of CD8+ $T_{EMRA-}$ and CD8+ $T_{EMRA+}$ cells during and after anti-OX40 administration. **f** Summary of the percentages of total CD19+ cells, Ki-67+ CD19+ cells and plasmablasts, identified by the absence of IgD and expression of CD38. **g** Expression of ICOS on CD4+ Tconv and Treg cells during and after OX40 treatment. All cells were gated excluding doublets and dead cells. Error bars indicate mean + SEM (**c–g**); *$P < 0.05$; **$P < 0.01$; ***$P < 0.001$; ****$P < 0.0001$; ns not significant. $P$ values were determined by paired two-tailed Student's $t$ test between D1 and D12 or D19. $N = 3$ patients in the D8 cohort, $N = 9$ patients in the D12 cohort, and $N = 4$ patients in the D19 cohort (**b–g**). Black triangles represent the D8, orange circles the D12, and blue squares the D19 cohort. The same color code is applied to the values denoting significance. Source data are provided as Source Data file.

lymphocytes was reduced marginally at D8 and D12, reflecting the lymphopenia that we observed, but was at baseline level by D19 after anti-OX40 administration (Fig. 1f, first panel). In parallel, albeit B-cell frequencies dropped slightly, Ki-67+ CD19 + cells were increased in the D12 cohort with a fivefold increase ($P = 0.0067$), and the CD38$^{hi}$ CD19+ B-cell subset (plasmablasts) increased tenfold by D12 ($P = 0.0107$) (Fig. 1f and Supplementary Fig. 1h). Together, these observations show evidence of significantly increased immune activation in most patients at D12 (and in some patients at D19) after anti-OX40, although increased B-cell proliferation could be due to initiation of a human anti-mouse antibody immune response (to MEDI6469), which was observed in our phase I study[24]. Lastly, expression of the activation marker inducible T-cell costimulatory (ICOS) protein was upregulated on CD4+ Tconv cells at D12 and D19 after anti-OX40 treatment ($P = 0.0066$, D12; $P = 0.0234$, D19) as well as on Treg cells in the D12 cohort ($P = 0.0085$, D12; $P = 0.2289$, D19) (Fig. 1g). These data indicate that anti-OX40 increased both activation and proliferation of peripheral CD4+ and CD8+ T cells 12 and 19 days after its administration in HNSCC patients.

**Anti-OX40 alters the composition of T-cell subsets within the tumor.** Diagnostic biopsies and surgical resection samples from 16 out of 17 patients were assessed for changes in TIL before and after anti-OX40 (Supplementary Fig. 3). Multicolor flow cytometry analysis revealed an increase in the percentage of CD8+ T cells with a concomitant decrease of CD4+ T cells in the TIL from 5 out of 16 patients after anti-OX40 treatment. Of note, Treg cells calculated as a percentage of total CD4+ T cells also increased after anti-OX40 in the same patients that showed increases in CD8+ T cells (Fig. 2a). Recently we reported that CD8+ TIL can be divided into three subsets based on expression of CD103 and CD39: CD103−CD39− double negative (DN), CD103+ CD39− single-positive (SP), and CD39+ CD103+ double-positive (DP) cells. DP CD8+ TIL are enriched for tumor reactivity, have a resident memory signature, and are present at low frequencies in the blood[27]. tSNE analysis of CD3+ cells within the TIL of pre- and post samples in a representative D12 patient, HNOX07, showed a distinct cluster of CD8+ cells that co-expressed CD103, CD39, and Ki-67, and this cluster was increased after anti-OX40 administration (Fig. 2b). Figure 2c shows a dot plot of CD103 and CD39 expression pre- and post anti-OX40 treatment on CD4+, CD8+, and Treg cells in the same patient (HNOX07). Supplementary Fig. 4a shows another patient with an increase in the DP CD8+ TIL subset after anti-OX40 treatment and a patient with low expression of this subset in the tumor, which remained unchanged after anti-OX40 treatment (Supplementary Fig. 4b, c). The expression of CD103 and CD39 in CD8+ TIL and CD39 in CD4+ TIL before and after

anti-OX40 for all patients is summarized in Fig. 2d. We also analyzed the expression of Ki-67 in the tumor before and after treatment. A representative patient (HNOX04) is shown in Fig. 2e where Ki-67+ CD4+ TIL were increased after treatment, while, among CD8+ TIL subsets, DP cells showed the highest increase in proliferation. A significant increase in Ki-67 expression was observed in the Tconv CD4+ TIL (11 of 16 patients). Increased CD8+ TIL proliferation post treatment was observed in 4 of 16 patients in the CD103+ CD39+ CD8+ TIL sub-population, while one patient showed an increase in Ki-67 expression in the SP subset.

**A CD8 TIL activation index to quantify immunological changes after anti-OX40.** In the anti-OX40 phase I study, we found a correlation between increased CD8+ T-cell proliferation and patients with regressing or stable disease[24]. In mouse models, we also described an increase in CD8+ TIL after anti-OX40 treatment[28], therefore we performed an in-depth analysis on CD8+ TIL before and after anti-OX40. Based on changes in the percentage of CD8+ TIL after anti-OX40 administration (increase in 5/16 patients), changes in CD103/CD39 expression on CD8+ TIL (increase in 8/16 patients), and proliferative changes in CD8+ TIL (Ki-67 expression increased in 4/16), we calculated an activation index based on the fold-change values comparing percentages at the DOS to baseline. All three categories combined were used to define patients with robust changes in CD8+ TIL (Supplementary Fig. 5a). Using these criteria, four patients showed robust activation in CD8 + TIL and were deemed "immunologic responders", two of which, HNOX04 and HNOX07, experienced a profound increase in this population post treatment (Fig. 3a). We also investigated whether the activation in the periphery would reflect increases in the tumor. Both, ICOS and Ki-67/CD38 were upregulated on peripheral Tconv CD4+ cells between D12 and D19 but did not segregate responders from non-responders. (Supplementary Fig. 5b). We believe that the increase in proliferating DP TIL represents robust activation of the tumor-reactive CD8+ TIL in 4 of 16 patients.

In order to examine whether anti-OX40 treatment affected the immune cell distribution in the tumor microenvironment, we analyzed FFPE patient samples pre- and post anti-OX40 treatment using mIHC. We first analyzed the expression of CD3, CD8, Foxp3, PD-L1, and CD163 to examine changes in T-cell subsets, PD-L1 expression, and presence of CD163+ macrophages after OX40 treatment. In line with published data[29], total CD3+, CD3+ Foxp3+, and CD3+ CD8+ T-cell numbers were significantly higher in the stroma of most HNSCC patients compared to the tumor, which was observed before and after anti-OX40 treatment (Fig. 3b and Supplementary Fig. 5c). We compared the number of CD163+ tumor-associated macrophages and PD-L1 expression in tumor versus stromal areas and found minimal changes in these subsets (Supplementary Fig. 3d).

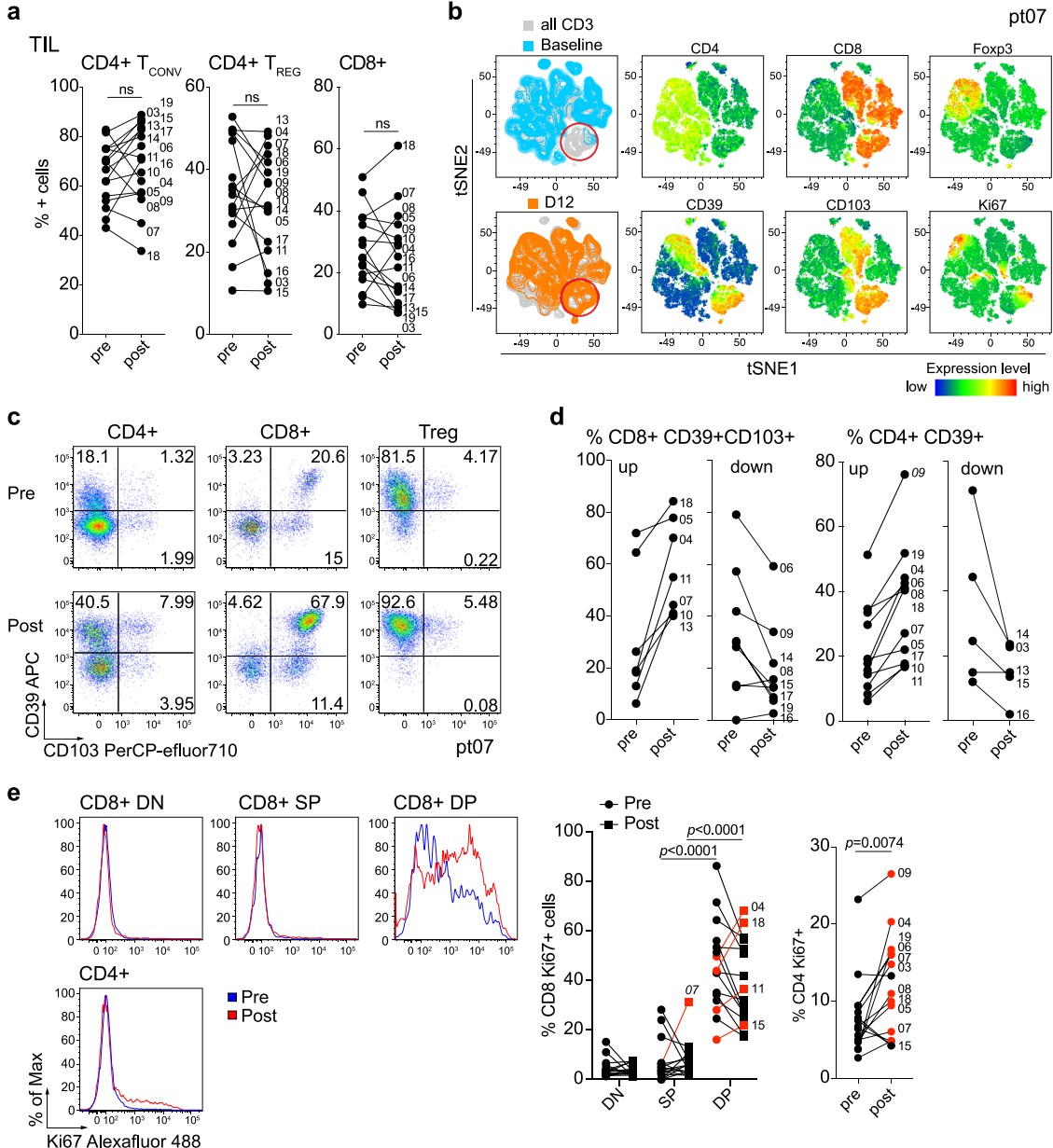

**Fig. 2 Changes in TIL composition after OX40 administration.** TIL from a pretreatment biopsy and a surgical specimen after OX40 therapy were assessed for lymphocyte composition and activation markers. The gating strategy is outlined in Supplementary Fig. 3. **a** Percentages of CD4+ Tconv cells, CD4+ Treg cells, and CD8+ T cells in each patient before and after OX40 administration, $N = 17$ patients. **b** tSNE analysis of the pre and post specimens from patient HNOX07, gated on CD3+ cells. Blue represents the baseline sample, orange the day of surgery sample and gray is the concatenated file. The red circle indicates the population of cells expressing both CD103 and CD39. tSNE analysis was performed on $N = 4$ patients, one representative patient is shown here, two more patients are shown in Supplementary Fig. 4. **c** Flow cytometric analysis of the expression of CD103 and CD39 in CD4+ Tconv cells, CD8+ cells, and CD4+ Treg cells in one immune-responding head and neck squamous cell carcinoma (HNSCC) patient pre- and post OX40 therapy. **d** Summary of the flow cytometric analysis in (c), left panel depicts CD8+ CD103+ CD39+ T cells and the right panel depicts CD4+ CD39+ T cells; patients with an increase are shown on the left, patients with a decrease are on the right. **e** Expression of Ki-67 was assessed among memory CD4+ TIL and CD8+ TIL subsets (DN, SP, and DP) in biopsy (pre) and DOS (post) tissue ($N = 17$ patients). Blue histograms indicate pre, red indicate post tissues. The left graph shows a summary of the percentage of proliferating CD8+ T-cell subsets pre- and post anti-OX40 in each patient. The graph on the right indicates the expression of Ki-67 in CD4+ TIL pre- and post anti-OX40. Red symbols highlight patients that exhibit an increase. *$P < 0.05$; **$P < 0.01$; ***$P < 0.001$; ****$P < 0.0001$; ns not significant. $P$ values were determined by paired two-tailed Student's $t$ test between pre- and post samples (**a**, **e**). Source data are provided as Source Data file.

Of note, the number of CD3+ cells expressing PD-L1 in the stroma was reduced markedly after anti-OX40 administration (Supplementary Fig. 5d). We then analyzed stroma and tumor in all patients for CD3, CD8, Ki-67, and CD103. CD39 was not assessed because there is currently not a suitable antibody for IHC staining. Figure 3c depicts tumor specimens from two

representative HPV+ patients; one patient (HNOX18, top) showed an increase in cells expressing CD8, CD103, and Ki-67 in the intratumoral areas after anti-OX40 treatment, whereas HNOX09 (bottom) showed no changes in stroma or tumor after treatment. Similar observations were made in HPV− patients; increased immune infiltrates were observed in some patients

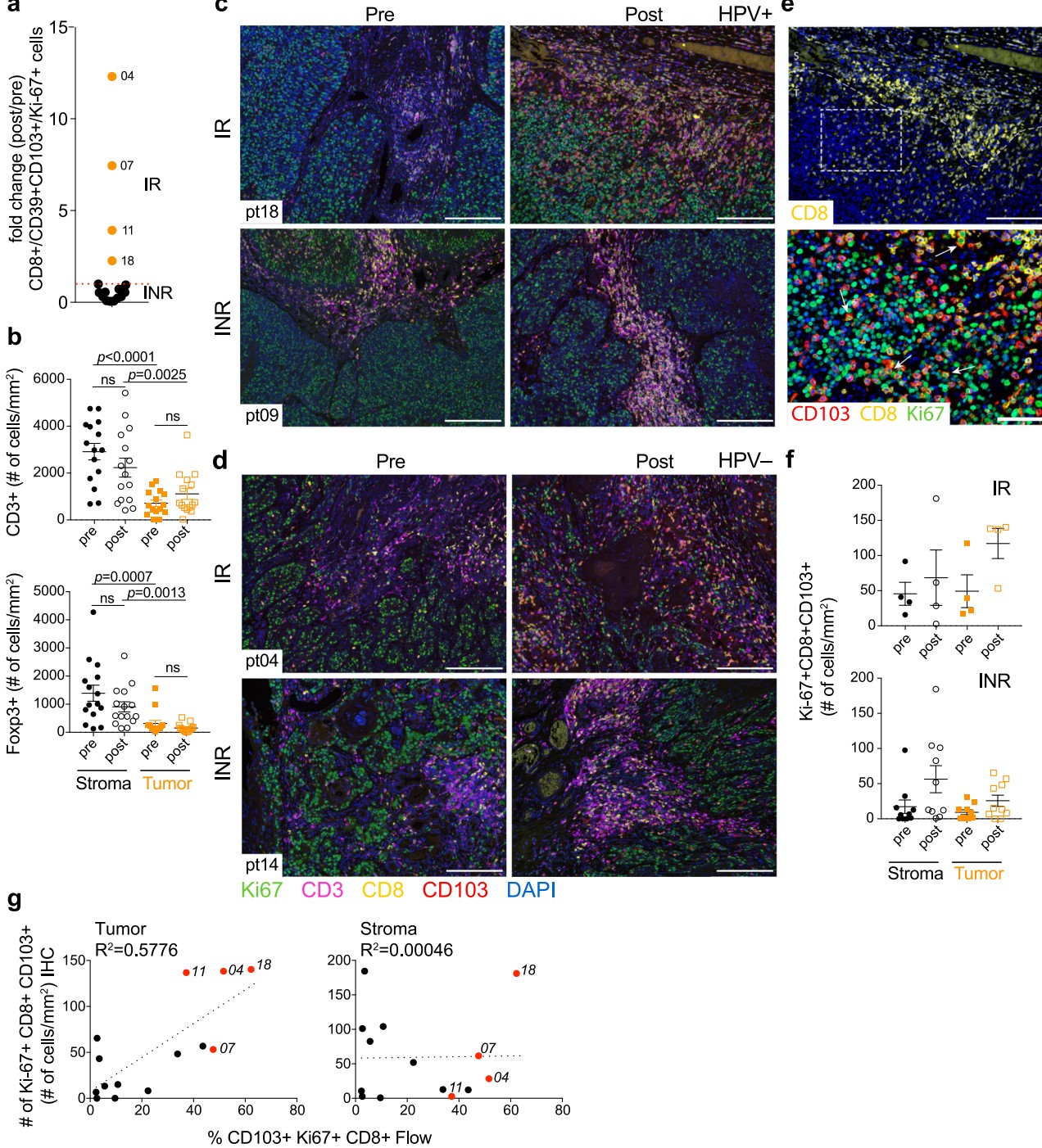

**Fig. 3 Multiplex IHC analysis reveals changes in lymphocyte infiltrates in tumor and stroma after anti-OX40.** Multiplex IHC was performed on FFPE specimens from $N = 15$ patients to determine the composition and changes of the immune infiltrate in tumor and stroma. For each patient, on each slide, six regions of interest (ROI) were analyzed for tumor and stroma. **a** Fold change of CD8+ CD103+ CD39+ Ki-67+ T cells for all patients as determined by flow cytometric analysis. Filled orange circles indicate patients with an activation index above 1. Filled black circles indicate patients with an activation index below 1. IR immune responder, INR immune non-responders. **b** Total number of CD3+ and Foxp3+ T cells among tumor and stroma in $N = 15$ patients pre- and post anti-OX40. Error bars indicate mean ± SEM, *$P < 0.05$; **$P < 0.01$; ***$P < 0.001$; ****$P < 0.0001$; ns not significant. $P$ values were determined by paired two-tailed Student's $t$ test between pre and post samples and between tumor and stroma. **c**, **d** Representative multiplex IHC of two HPV+ (**c**) and two HPV− (**d**) head and neck squamous cell carcinoma (HNSCC) patients pre and post anti-OX40, identified by the response (IR and INR) as determined in (**a**). **e** Magnified view (bottom) of cells expressing CD8, CD103, and Ki-67 in the tumor area (top). **f** Summary of the numbers of Ki-67+ CD103+ CD8+ cells in the tumor and stroma pre- and post anti-OX40, in patients stratified by the immunologic response. $N = 4$ IR, $N = 10$ INR. Error bars indicate mean ± SEM. The values presented in (**b**) and (**f**) represent the mean of the ROI analysis for each patient. **g** Comparison of the numbers of Ki-67+ CD103+ CD8+ cells enumerated by multiplex IHC in the tumor and stroma, with the percentages of the same subset determined by flow cytometry. Black dots represent the immune non-responders. Red dots represent the immune responders. Source data are provided as Source Data file.

(HNOX04, top); with little to no change in other patients after anti-OX40 (HNOX14, bottom) (Fig. 3d). Figure 3e is a high-power image of T cells expressing CD8, CD103, and Ki-67 within the tumor, which we believe represent the proliferating CD103+ CD39+ CD8+ TIL. When flow cytometry data were analyzed after grouping patients according to immunologic responders (IR) versus non-responders (INR) (Fig. 3a), we observed that the increase in proliferating Ki-67+ CD103+ CD8+ CD3+ T cells was largely confined to the intratumoral compartment rather than the stroma in the four responding patients (tumor: 3.4-fold increase ±2.04, vs stroma: 1.12 ± 0.78) (Fig. 3f and Supplementary Fig. 5e). Finally, we compared whether the frequencies of DP CD8+ TIL obtained by flow cytometry correlated with observations by IHC and if so, whether there was any difference between IHC data obtained from within the tumor or stroma. The flow cytometry data correlated well with cell counts obtained from intratumor areas ($R^2 = 0.5776$), but not as well with cell counts in the stroma ($R^2 = 0.00046$) (Fig. 3g). Together, these data suggest that in some patients CD103+ CD39+ CD8+ TIL are preferentially expanded and/or recruited into the tumor areas after anti-OX40, potentially limiting tumor growth.

**Survival analysis after neoadjuvant anti-OX40**. The median follow-up time for this group of patients is 39 months (95% CI: 34–45 months)[30]. The overall and disease-free survival for the entire study population was 94 and 71% at 1.5 years and 82 and 71% at 3 years, respectively (Fig. 4a Supplementary Table 4a). Figure 4a and Supplementary Table 4b indicate the number of patients at risk for the Kaplan–Meier curve[31]. Five of 17 patients have recurred, two of which remain alive with stable disease, and are currently being treated with various immunotherapy and conventional treatment regimens. When we segregated patients into immunologic responders and non-responders as defined in Fig. 3a, it was noted that all four immunologic responders are alive without recurrence, despite having the high-risk disease (3/4 immunologic responders were HPV−) (Fig. 4b). Of the 6 HPV+ patients, only one has recurred, whereas 4 of 11 (36%) HPV− patients recurred after surgery, which is lower than the expected recurrence rate in this population of locoregionally advanced patients (50–65%)[32].

**Effects of anti-OX40 on T-cell clones in tumor and blood**. To better understand the molecular characteristics of T cells in the tumor after anti-OX40 administration, we performed high-throughput TCRβ sequencing analysis on selected patients, where sufficient tissue was available. In other studies, T-cell receptor repertoires were assessed in total TIL populations, often isolated from FFPE samples, which does not distinguish the source of the TCR sequences found within the tumor (e.g., Tconv CD4+, Treg CD4+, CD8+, or natural killer T cells). We analyzed the TCRβ repertoire in both blood (pre- and post anti-OX40) and TIL isolated from tumor digest from four patients (biopsy tissue was examined in two out of the four patients). Memory CD4+ Tconv cells and memory CD8+ T-cell subsets were isolated by flow cytometric cell sorting and the TCRβ CDR3 region was sequenced (Fig. 5a and Supplementary Fig. 6a, b). First, the clonality of CD4+ and CD8+ T cells in blood and matching drLN samples pre- and post anti-OX40 was analyzed in these patients. A clonality score of 1 indicates a monoclonal population, whereas clonality values close to 0 indicate very diverse, polyclonal populations. Peripheral CD4+ T cells from HNOX05 and HNOX11 displayed a small increase in clonality, whereas clonality decreased in HNOX04 and HNOX18 post OX40. Clonality in the peripheral CD8+ T-cell compartment increased in three out of the four patients after anti-OX40

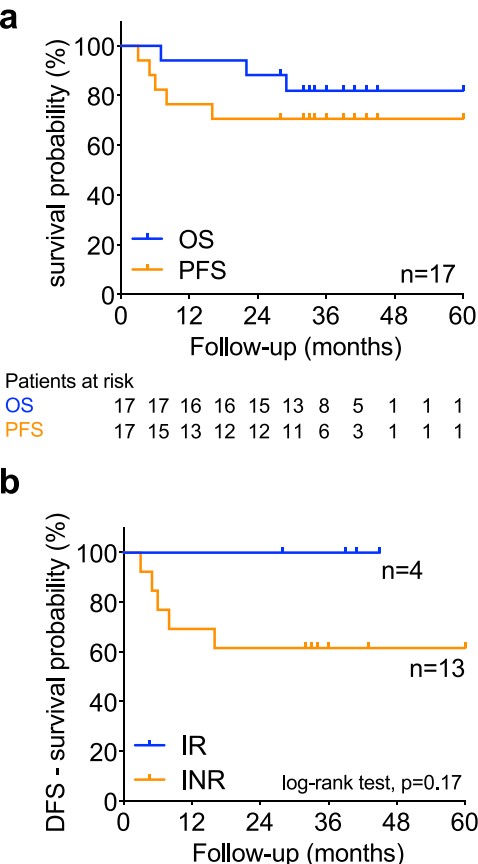

**Fig. 4 Survival analysis of patients after neoadjuvant anti-OX40 administration.** Patients were followed for disease-free (DFS) and overall survival (OS). **a** Kaplan–Meier estimate of DFS (orange) and OS (blue). The numbers below the graph represent the number of patients at risk. $N = 17$ head and neck squamous cell carcinoma (HNSCC) patients. Supplementary Table 4b contains the survival estimate for the overall and disease-free survival dataset as outlined in Gebski et al.[31]. **b** DFS in immune-responding ($N = 4$, blue) vs non-responding ($N = 13$, orange) HNSCC patients. The log-rank test (Mantel–Cox) was used to compare both curves and P values < 0.05 were considered statistically significant. Source data are provided as Source Data file.

treatment (Fig. 5b). Of note, CD4+ T-cell clonality in the TIL increased twofold after anti-OX40 treatment; however, in DN, SP, and DP CD8+ subsets clonality was unchanged (Fig. 5c), which could be due to an influx of other CD8+ T-cells clones, thus diluting clones that expanded in situ. To compare the distribution of CD4+ and CD8+ T-cell clones within the tumor and periphery, we used the Morisita–Horn index, which calculates clonal overlap between two populations. TCRβ CDR3 sequences of sorted peripheral memory CD8+ T cells were isolated from pre- and post anti-OX40 samples. There was a significant overlap of memory CD8+ T cells with both the DN and SP subsets but much less with the DP CD8+ TIL (Supplementary Fig. 7a). By focusing our analysis on the TIL subsets and drLN post treatment, we found that TCR sequences from DN and SP subsets overlap significantly in all four patients tested and share sequences with the drLN CD8+ T cells (Fig. 5d, left and middle panel). In contrast, DP TIL had a unique TCRβ repertoire (no overlap between DN, SP, or drLN CD8+ cells) which may have been the result of local intratumoral expansion, as suggested by increased Ki-67 levels (Figs. 3e and 5d)[27]. To examine this further, we evaluated the clonality within the DP CD8+ TIL population and found that the top 30 clones accounted for

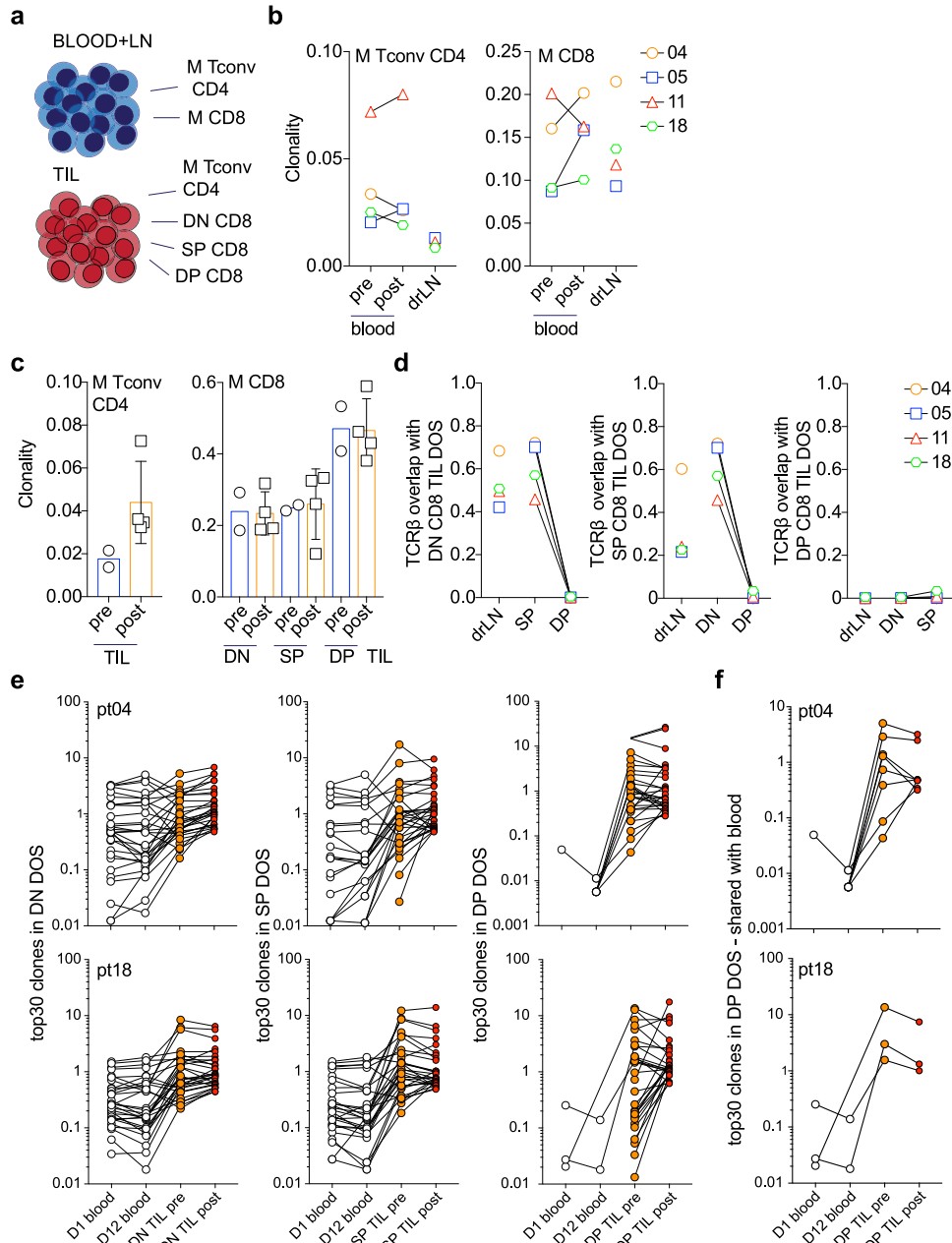

**Fig. 5 TCRβ sequencing analysis reveals clonal differences following administration of anti-OX40.** TCRβ sequencing was performed on blood and TIL samples obtained before and after anti-OX40 administration. The gating strategy for the isolation of the blood and TIL subsets is outlined in Supplementary Fig. 6a, b. **a** Schematic of all T-cell subsets that were isolated from blood and tumor by cell sorting, followed by DNA isolation and TCRβ sequencing. **b** Summary of the clonality of CD4+ and CD8+ T cells in blood before anti-OX40 as well as blood and drLN at DOS in four patients. **c** Summary of the clonality of CD4+ TIL and CD8+ TIL subsets (DN, SP, and DP) before (pre) and after (post) anti-OX40. $N = 2$ pre, $N = 4$ post. Error bars indicate mean ± SEM in the post patients. **d** TCRβ repertoire overlap was calculated using the Morisita–Horn index. Overlap analysis is shown for DN CD8+ TIL compared with SP, DP, and drLN cells. The same analysis was performed for SP CD8+ TIL (with DN, DP, and drLN) and DP CD8+ TIL (with DN, SP, and drLN). Colors and symbols in (**b**) and (**d**) depict the four patients. **e** The top 30 clones in the DOS specimen (DN, SP, and DP) were separately compared to the same subsets pretreatment and to memory CD8+ T cells in blood at D1 and D12. Black open circles represent the blood before and after anti-OX40, orange filled circles represent the biopsy (pre) specimen, and red filled circles the DOS (post) specimen. Connecting lines indicate the presence of the same TCRβ sequence in each subset. **f** Red filled circles represent the number of clones among the top 30 clones in DP CD8+ TIL post treatment, that were present in the sample pretreatment. In **b**–**f**, $N = 4$ patients (HNOX04, HNOX05, HNOX11 and HNOX18) were analyzed. M memory, LN lymph node, DOS day of surgery.

76–96% of all CDR3 sequences, with the top three clones present at frequencies above 30% (Supplementary Fig. 7b).

To better understand the relationship of T cells in the blood and tumor before and after anti-OX40 treatment, we focused on those top 30 clones in DN, SP, and DP CD8+ TIL subsets post

treatment and asked whether they were present in pretreatment samples. Figure 5e separately depicts DN, SP, and DP subsets from HNOX04 and HNOX18, and the frequencies of shared clones in blood and tumor at baseline and DOS. In all subsets, the frequency of clones in TIL is higher than in blood, and some

clones increase or decrease differentially after anti-OX40 treatment. However, among the top 30 clones found in the DP TIL post treatment, only eight and three clones were shared with the peripheral T cells in HNOX04 and HNOX18, respectively (Fig. 5f). Similar results were obtained for HNOX05 and HNOX11 (three and eight clones, respectively) (Supplementary Fig. 7c).

Lastly, we sought to determine whether the CD8+ TIL recognized tumor antigens in patients that showed an immunologic response as assessed by the activation index in Fig. 3a. We sorted and expanded DN, SP, and DP CD8+ T cells from HNOX04, HNOX07, and HNOX18 (all IR) and screened them for neoantigen reactivity after predicting neoantigens based on whole-exome and RNA-sequencing data. For HNOX18 (HPV+ patient), we did not detect reactivity of DN, SP, or DP CD8+ T cells to any of the predicted 133 mutated peptides we screened (Supplementary Data 2). However, when DN, SP, and DP cells from HNOX18 were screened against peptides from HPV16 and HPV18 E6 and E7 proteins (tumor-associated antigens), we observed a strong response to HPV16 E6 and E7 only in the DP CD8+ T-cell subset (Fig. 6a, b). To investigate whether the HPV-specific response was due to one or several T-cell clones, we sorted 4-1BB-CD25- (non-reactive) T cells and 4-1BB+ CD25+ (reactive) T cells after 18 h coculture with HPV16 E6 and E7 transfected PBMC. Response to E6 was driven by four dominant T-cell clones and response to E7 comprised two dominant T-cell clones (clonality cutoff >2%) (Fig. 6c). For HNOX04 and HNOX07 (HPV− IR), 85 and 29 potentially immunogenic neoantigen peptides were predicted, respectively (Supplementary Data 2). Reactivity to one unique neoantigen for each patient was detected (Fig. 6d–g). Reactivity to a mutated peptide from SP100 (a tumor suppressor gene) was restricted to the DP CD8+ TIL (Fig. 6d, e; HNOX04), whereas in HNOX07 there was reactivity in both SP and DP cells to mutated PPP1R13L (inhibitor of p53) (Fig. 6f, g). Together these data confirm that tumor-antigen-specific T cells are enriched within the DP CD8+ TIL subset in HNSCC patients. Since anti-OX40 was able to increase the frequency of the DP CD8+ TIL subset in the immunologic responder patients, we hypothesize that the increased frequency of the tumor-reactive T cells after treatment aids in tumor clearance and protection from recurrence (as demonstrated by survival benefit in Fig. 4a, b).

## Discussion

Here, we describe the results from a first-in-human neoadjuvant clinical trial treating HNSCC patients with an OX40 agonist antibody and the subsequent immunologic changes in tumors and peripheral blood after drug infusion. Administration of anti-OX40 (MEDI6469) prior to surgery was safe, did not delay surgery, and resulted in no unexpected surgical complications. We observed immune activation illustrated by increased CD4+ and CD8+ T-cell proliferation following treatment, which peaked between 2 and 3 weeks after antibody infusion. Importantly, we also demonstrated that anti-OX40 increased clonality in the peripheral blood CD8+ T-cell compartment and clonality of tumor-infiltrating CD4+ cells of some patients. Expansion of tumor-antigen-specific CD103+ CD39+ CD8+ TIL was observed in 4 of 16 patients with evaluable pre- and post-treatment samples. Understanding why the expansion of the DP CD8+ TIL occurred in only 25% of patients will require further investigation, but may be related to the presence of negative regulatory elements within the tumor (e.g., Treg cells, myeloid-derived suppressor cells, TGF-β, expression of checkpoint ligands). Hence, blocking negative immune regulators or factors while boosting OX40 signaling may lead to greater CD8+ TIL

activation in a larger fraction of patients. Despite considerable enthusiasm for anti-OX40 based on our initial phase I trial[24], humanized OX40 agonists have not exhibited significant clinical activity when used as monotherapy or in combination with checkpoint inhibition. Lack of efficacy might be related to schedule and dosing[33,34]. While adequate for checkpoint proteins, dosing 2–3 weeks apart may not be optimal for agonist antibodies. Of note, preclinical data and data in our clinical trials demonstrated immune activation and tumor reduction after short pulses of anti-OX40 (given 2–3×/week)[24,35].

Much work has been devoted to the analysis and characterization of the treatment-naive tumor microenvironment, so as to inform patient care and possibly a response to therapy. However, obtaining study material can be challenging, and therefore detection of biomarkers in peripheral blood to predict response would be ideal. Immune activation in peripheral CD4+ and CD8+ T cells peaked between D12 and D19 after anti-OX40 and these cells were characterized by high expression of ICOS, Ki-67, and CD38. Patients in the D8 cohort might have exhibited a similar increase in peripheral blood immune activation at a later time, however, the trial was not designed to include additional blood draws at 4- or 11-days post surgery. In a future study, additional timepoints will be included. The expansion cohort was informed based on increased proliferation between D12 and D19 and D12+/− 2-day patients were accrued for the remainder of the trial, with the exception of patient HNOX16, whose surgery was delayed and thus included in the D19 cohort. Data from the Sharma laboratory in both preclinical models as well as patients with bladder cancer, prostate cancer, and metastatic melanoma suggested that increased ICOS protein expression on CD4+ T cells was indicative of a response to CTLA-4 treatment[10,36–39]. In our patients, ICOS upregulation on peripheral CD4+ cells did not distinguish immune responders and non-responders (Supplementary Fig. 5b). Hence, we believe that ICOS alone as a biomarker on CD4+ cells is not sufficient to differentiate robust immune responses to anti-OX40; however, a combination of ICOS with an activation marker, such as 4-1BB or CD40L, may help segregate patients into immunologic responders versus non-responders. In future studies, high-dimensional multi-parameter blood profiling using CYTOF and/or state-of-the-art flow cytometry approaches might aid in finding immune-based biomarkers of response.

Preclinical data from our laboratory has shown that anti-OX40 was superior to CTLA-4[28] in its ability to enhance the proliferation and survival of vaccine-stimulated T cells. We and others have also demonstrated that anti-OX40 can reinvigorate exhausted/anergic T cells and promote priming and maintenance of tumor-antigen-specific CD8+ cells in the periphery[40,41]. In line with this observation, Crittenden et al. showed in a mouse model that tumor-activated T cells generated after tumor implantation are critical for the T-cell response subsequent to radiation and checkpoint inhibition[42]. Based on these data, and results from our trial, we hypothesize, that anti-OX40 not only increases the priming of peripheral T cells to tumor antigens but also boosts pre-existing tumor-antigen-specific T cells. When followed by PD-1 blockade, this combination could limit T-cell exhaustion within the tumor, resulting in improved tumor control. Several laboratories, including ours, have addressed this critical question in preclinical studies and, while the sequence of administration and dosing appears to be dependent on respective mouse models, combined treatment of anti-OX40 with checkpoint blockade has shown greater anti-tumor activity[33,34,43].

In the tumor, we focused our analysis on a population of tumor-resident CD8+ T cells that we have shown are highly enriched for tumor–antigen reactivity[27]. With access to tumor specimens before and after anti-OX40, we found an increase in CD8+ CD103+ CD39+ cells in the tumor in 4 of 16 patients

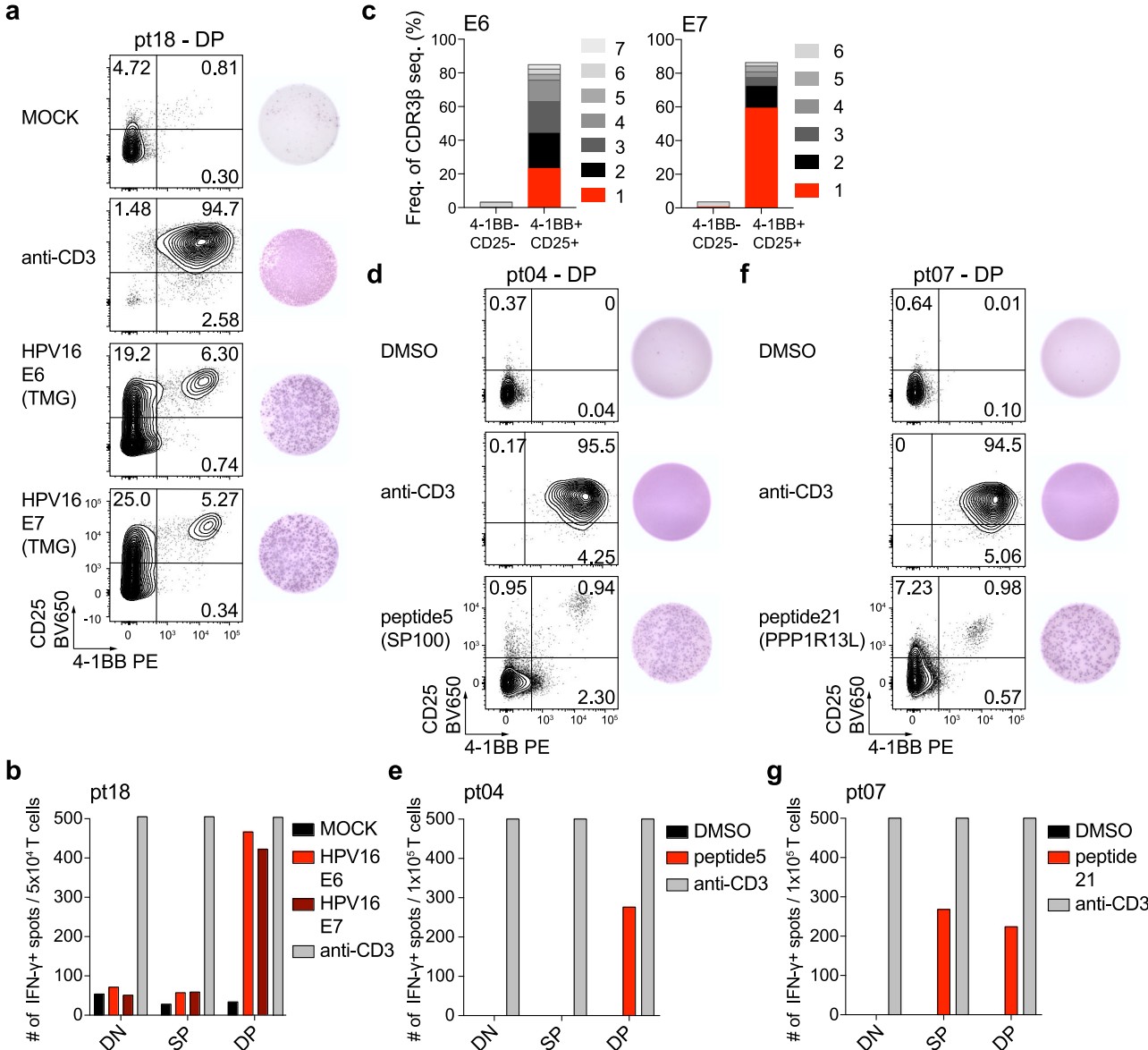

**Fig. 6 Identification of neoantigen and HPV-reactive cells in patients after anti-OX40.** Expanded DP CD8+ T cells from HNOX18 were screened for reactivity against HPV16, and neoantigens were predicted for HNOX04 and HNOX07. Peptide Pools were screened for reactivity by IFN-γ -ELISpot analysis. **a** Expanded DP CD8+ T cells from HNOX18 were screened with autologous PBMC transfected by electroporation with RNA encoding HPV16 E6 and E7 proteins. Anti-CD3 is a positive control, water the negative (MOCK) control. Prior to IFN-γ ELISpot development (right), cells were harvested and expression of 4-1BB and CD25 was assessed by flow cytometry (left). **b** Summary of HPV-specific spot-forming cells (SFC) in CD8+ TIL subsets. **c** HPV16 E6 and HPV16 E7 reactive CD8+ T cells were sorted based on the expression of 4-1BB and CD25. 4-1BB-CD25− and 4-1BB+ CD25+ cells were analyzed by TCRβ-sequencing and the frequency of the top 6 (E6) and top 7 (E7) clones are depicted. **d** Expanded DP CD8+ T cells from patient HNOX04 were screened with the addition of predicted neoantigens. Shown is the response to peptide 5, anti-CD3 as positive, DMSO as the negative control. **e** Summary of spot-forming cells (SFC) in CD8+ TIL subsets from HNOX04. **f** Expanded DP T cells from HNOX07 were screened with the addition of predicted neoantigens. Shown is the response to peptide 21, anti-CD3 as positive, DMSO as the negative control. **g** Summary of SFC in CD8+ TIL subsets from HNOX07. Red depicts the peptide response, gray is the anti-CD3 control, black the MOCK/DMSO control. TMG tandem minigene. Source data are provided as Source Data file.

(3 HPV−, 1 HPV+) after anti-OX40 infusion and none of these patients have had a tumor recurrence. Twelve of 16 patients (7 HPV−, 5 HPV+) displayed less dramatic changes within the DP CD8+ TIL and 5 have recurred thus far (with one death). Therefore, seven patients that failed to exhibit an increase in our activation index are still tumor-free. This could be due to timing—D8 patient samples (blood and TIL) were taken prior to the peak in immune activation—or the tumor microenvironment in these tumors was more suppressive and prevented infiltration and/or expansion of CD103+ CD39+ CD8+ TIL.

TCR diversity, which is often used as a surrogate for immune competence, has been investigated in many immune-oncology clinical trials. We analyzed the T-cell repertoire in sorted T-cell subsets from DOS specimens and blood before and after anti-OX40 treatment in four patients, two of whom we also obtained a biopsy prior to anti-OX40 dosing. Using sorted cells allowed us to discriminate CD4+ Tconv, Treg, and CD8+ cells and more specifically, to distinguish tumor-specific cells (DP) from bystander cells (DN, SP) which are abundant in some tumors[44]. T-cell clones in both patients increased up to 15-fold after

anti-OX40; however, the DP subset exhibited the highest and most robust increases compared to DN and SP cells (Fig. 5f). To address whether DP CD8+ TIL were enriched for tumor- or tumor-associated antigen reactivity, we tested the reactivity of DN, SP, and DP TIL to neoantigens for three patients. While an HPV-specific and a neoantigen-specific T-cell response were found in 2/3 patients exclusively within the DP TIL population, the reactivity for one neoantigen was also found in both the DP and SP CD8 TIL in one patient (HNOX07). These tumor antigen-specific T cells within the CD103 SP TIL may not have upregulated CD39 and/or expressed lower levels of CD39 at the time of surgery. Based on our previous publication[27], we also believe that the high-frequency clones (Supplementary Fig. 7b) within the DP TIL fraction in HPV− patients are most likely tumor antigen-specific, but were not detected as such using our mutated short peptide/MHC I binding algorithm approach. In collaboration with a group in the Netherlands, we showed that in colon cancer patients, reactivity to neoantigens was confined to the CD103+ CD39+ CD8+ TIL[45]. Recent studies have highlighted limitations in current mutated antigen pipelines suggesting that intronic regions could contribute to the neoantigen pool[46] and some of the T-cell responses could be directed to overexpressed self-antigens. Furthermore, depending on the tumor type and the intrinsic tumor-reactivity of the intratumoral TCR repertoire, the capacity to recognize autologous tumor can vary substantially[47] and those biologic and technical limitations will be addressed in future studies.

In summary, we have shown that anti-OX40 delivered in the neoadjuvant setting has few side effects and does not delay surgical resection in HNSCC patients. Extensive immune monitoring demonstrated that the majority of patients exhibited evidence of increased peripheral blood CD4+ and CD8+ T-cell activation and proliferation, with a subset of patients showing the expansion of CD103+ CD39+ CD8+ TIL, which are enriched for tumor-reactivity, and correlate with greater disease-free survival. Our group has recently initiated a clinical trial testing a humanized OX40 agonist (MEDI0562) in the neoadjuvant setting in HNSCC and melanoma patients, which will allow us to assess the agonist activity of an anti-OX40 antibody independent of anti-drug immune responses. In parallel, we are investigating multiple factors that may limit immune responses in cancer patients treated with anti-OX40, which should reveal ways to enhance tumor antigen-specific T-cell function in a greater number of patients.

## Methods

**Patients.** Eligible patients were 18 years of age or older and had stage II, III, or IVA HNSCC that was considered surgically resectable. All patients had an Eastern Cooperative Oncology Group (ECOG) performance-status score of 0 or 1. Exclusion criteria were immunodeficiency, ongoing systemic immunosuppressive therapy, active autoimmune or infectious disease, clinically significant concurrent cancer, and medical or psychiatric condition that in the opinion of the PI would preclude compliance with study procedures.

**Study design.** This two-stage study (PH&S IRB # 14-042, NCT02274155) started with a Phase Ib time interval reduction design with a constant anti-OX40 dose based on the previous Phase I trial. The stage I portion of the study was to include at least nine patients with a six patient safety run. The second stage would include an expansion cohort of up to 21 additional patients in the cohort selected, based on the most promising immune response in peripheral blood and within tumors of patients enrolled in stage I. No statistical tests were performed to calculate the sample size. Eligible patients were registered and provided written informed consent, after which they were assigned to a cohort based upon practical scheduling and patient desire (no randomization):

Cohort 1: anti-OX40 0.4 mg/kg IV d1,3,5, surgery 2 weeks later (d17–21)
Cohort 2: anti-OX40 0.4 mg/kg IV d1,3,5, surgery 1 week later (d10–14)
Cohort 3: anti-OX40 0.4 mg/kg IV d1,3,5, surgery 2–3 days later (d7–8)

The primary endpoint was a delay in time to surgery; secondary endpoints included exploratory assessments to determine the timing of peak immunologic effect and to compare the composition and immunologic phenotypes of TIL subsets before and after anti-OX40 administration. The first patient was enrolled on December 5, 2015 (HNOX01), and the last patient was enrolled on April 17, 2017 (HNOX19). All patients were monitored for adverse events, according to the National Cancer Institute Common Terminology Criteria for Adverse Events, version 4.0. Patients were deemed ineligible if planned surgery was delayed more than 3 days. All patients underwent the following: baseline tumor staging with a contrast-enhanced CT or magnetic resonance imaging of the neck and chest; positron-emission tomography–computed tomography (PET–CT); and a pretreatment biopsy of the primary tumor or metastatic lymph node, a portion of which was harvested as a fresh research specimen for flow cytometric analysis. Resection of the primary tumor, metastatic, and draining lymph nodes was completed as part of the standard of care and appropriate microvascular reconstruction was performed as indicated. All patients were offered conventional adjuvant risk-adapted radiotherapy or chemoradiotherapy per NCCN guidelines and were followed for recurrence-free and overall survival.

**Clinical assessments.** Anti-OX40 was infused at 0.4 mg/kg IV over 60 min for three doses, 48–96 h apart. The timing of surgery and number of assessments between anti-OX40 and surgery depended on cohort assignment. Repeat neck CT to assess changes in tumor volume for surgical planning purposes was performed pre-operatively at the discretion of the treating physician. Peripheral blood draws to assess immune parameters were mandatory on day 1, day 12+/− 2d, day 34+/− 3d, and day 55+/− 7d of the study. A research nurse performed an autoimmune disease assessment on day 1, day 12+/− 2d, day 34+/− 3d, and day 55+/− 7d. Pregnancy testing with b-HCG was mandatory on day 1 of treatment for pre-menopausal females. Follow-up after day 55+/− 7d (the completion of the study) was dictated by the clinical response. Patients had a physical exam, interval imaging every 3 months and for the first 2 years following surgical treatment, and every 6 months thereafter per treating MD recommendations. Another treatment as deemed medically appropriate was offered to patients with disease progression. All survival endpoints for immunologic responders versus non-responders were calculated from the day of surgery.

**Trial oversight.** This study was approved by the institutional review board at Providence Health and Services-Oregon. The study was designed, and the paper was written by the authors, who are responsible for the accuracy of its content. The trial was conducted in accordance with the ethical principles of the Declaration of Helsinki and with adherence to the Good Clinical Practice guidelines, as defined by the International Conference on Harmonization. The study also complies with the ICMJE guidelines on reporting. MEDI6469 (clone 9B12) was supplied by Med-Immune; 9B12 is a murine IgG1, anti-OX40 monoclonal antibody (mAb) directed against the extracellular domain of human OX40 (CD134) and was administered at 0.4 mg/kg on days 1, 3, and 5. The company did not have another role in the study or the report.

**Survival data and statistical analysis.** Overall- and disease-free survival were calculated from the date of surgery until last contact or death or disease recurrence, respectively, using the Kaplan–Meier method. The reverse Kaplan–Meier method was used to calculate median follow-up[30]. The number at risk was presented as a supplementary table for overall and disease-free survival as outlined in Gebski et al.[31]. Immunologic, histologic, and genomic analyses were performed on selected/available biospecimens, and correlative data were analyzed as described in the respective sections. A generalized estimating equation (GEE) model[48,49] was utilized to compare measurements over time in order to account for correlation among repeated measures from the same subject. Correlation structures including exchangeable, autoregressive with order 1 or AR (1), and independence were evaluated using quasi-likelihood under the independence model for the criterion (QIC) for analysis. The exchangeable model was selected for the GEE and reported in Supplementary Data 3. Statistical significance between baseline and D12 or D19 was determined by paired Student's $t$ test or the mixed-effects model and Tukey multiple comparison test using GraphPad Prism 8 software (GraphPad, San Diego, CA). Reported $P$ values are two-sided, and the significance level was set at 0.05 for all analyses unless otherwise noted.

**Patient samples.** Peripheral blood, uninvolved LNs, metastatic LNs, and tumor samples were obtained from all HNSCC patients. All subjects signed written informed consent approved by the Providence Portland Medical Center Institutional Review Board (PH&S IRB # 14-042). At the time of enrollment, patients were not undergoing therapy. PBMC were purified from whole blood over Ficoll-Paque PLUS (GE Healthcare) gradient and cryopreserved prior to analysis. Tumor specimens were prepared as follows: Under sterile conditions, tumors were cut into small pieces and digested in RPMI-1640 supplemented with hyaluronidase at 0.5 mg/ml, collagenase at 1 mg/ml (both Sigma-Aldrich), DNase at 30 U/ml (Roche) as well as human serum albumin (MP Biomedicals) at 1.5% final concentration. Cells were digested for 1 h at room temperature under agitation with a magnetic stir bar. Cell suspensions were filtered through a 70-μm filter. TIL were

enriched as described above by Ficoll-Paque PLUS density centrifugation. Tumor single-cell suspensions were cryopreserved in LN₂ until further analysis.

**Antibodies and flow cytometry**. Fresh blood and cryopreserved samples were used for flow cytometry studies.

The following fluorescent-labeled antibodies were used in various combinations: allophycocyanin (APC)-Cy7 and brilliant violet (BV) 605 anti-CD3 (UCHT1; 1:100—#300426 and #300406, respectively), BV785 anti-CD4 (OKT-4; 1:200—#317442), BV510 anti-CD8 (RPA-T8; 1:100—#301048), PerCP/Cy5.5 anti-CD19 (HIB19; 1:50—#302230), BV650 anti-CD25 (BC96; 1:100—#302634), APC and Alexa Fluor (AF) 488 anti-CD38 (HIT2; 1:50—#303510 and #303512, respectively), AF700 and BV711 anti-CD45RA (HI100; 1:50—#304120 and 304137), BV605 anti-CD69 (FN50; 1:50—#310938), BV421 anti-CD127 (A019D5; 1:50—#351310), PE-Cy7 anti-PD-1 (EH12.2H7; 1:50—#329918),PE-Cy7 anti-4-1BB (4B4-1; 1:40—#309818), PE/Dazzle 594 anti-CCR7 (G043H7; 1:50—#353236), BV711 anti-HLA-DR (L243; 1:100—#307644), BV510 anti-IgD (IA6-2; 1:50—#348220) (all from Biolegend); PE-CF594 anti-CD24 (ML5; 1:50—#562405), BV421 anti-CD27 (M-T271; 1:50—#562513), FITC anti-CD127 (HIL-7R-M21; 1:10—#560549), PE anti-OX40 (ACT35; 1:40—#555838), PE-Cy7 anti-PD-1 (EH12.1; 1:50—#561272), PE anti-granzyme B (GB11; 1:200—#561142), AF 488 and PE anti-Ki-67 (B56; 1:50—#561166 and 1:50 #556027) (all from BD Biosciences); APC-efluor780 anti-CD8 (RPA-T8; 1:100—#47-0088-42), APC and PE-Cy7 anti-CD39 (eBioA1; 1:100—#17-0399-42 and #25-0399-42, respectively), PE and PerCP-efluor710 anti-CD103 (B-Ly7 and Ber-ACT8; 1:100—#12-1038-42 and 1:50—#46-1037-42, respectively), efluor450 and AF700 anti-Foxp3 (PCH101; 1:40—#48-4776-41 and 1:25—#56-4776-41, respectively), biotinylated anti-ICOS (ISA-3; 1:100—#13-9948-82), Streptavidin APC-efluor780 (1:100—#47-4317-82) (all from eBioscience); A fixable live/dead dye was used to distinguish viable cells (Biolegend). Cell surface staining was performed in FACS buffer (PBS, supplemented with 1% FBS and 0.01% NaN₃). Intranuclear staining was performed using the Fix/Perm kit from eBioscience according to the manufacturer's instructions.

Stained cells were acquired on an LSRII and LSRFortessa flow cytometer, the FACS AriaII (all BD Biosciences) for cell sorting, and the Attune Nxt flow cytometer (Thermo Fisher Scientific). Data were acquired using FACS Diva and the Attune Nxt flow cytometer software, respectively. Data were analyzed with FlowJo software (Treestar).

**Cell sorting and T-cell expansion**. Cryopreserved PBMC and TIL were thawed and enriched for T lymphocytes using the T-cell enrichment kit from Stemcell for ex vivo staining. For TIL enrichment, EpCAM beads (StemCell) were added to the cocktail. The enriched fractions were then labeled and populations of interest were purified after cell sorting to 99% purity on a FACSAria II. Briefly, naïve CD8+ T cells were sorted as CD8+CD4−CD45RA+CCR7+ cells and memory CD8+ T cells were sorted as CD8+CD4−CD45RA−CCR7+/− (total memory). Naïve CD4+ subsets were sorted as CD4+ CD8−CD45RA+CCR7+ Foxp3-CD25- cells and memory CD4+ T cells in blood and PBMC were sorted as CD4+ CD8−CD45RA−CCR7+/− Foxp3- (total memory). CD8+ subsets from TIL were sorted as CD3+CD4−CD8+CD45RA−CCR7+/−CD39−CD103−(DN), CD3+ CD4−CD8+CD45RA−CCR7+/−CD39−CD103+ (SP), and CD3+CD4−CD8+ CD45RA−CCR7+/−CD39+CD103+ (DP). For TCR sequencing analysis, cell pellets were frozen after cell sorting until further processing.

For the expansion of DN, SP, and DP CD8+ TILs as well as naïve and memory CD8+ T cells, cells were sorted and cultured in complete RPMI-1640, supplemented with 2 mM glutamine, 1% (vol/vol) nonessential amino acids, 1% (vol/vol) sodium pyruvate, penicillin (50 U/ml), streptomycin (50 μg/ml), and 10% fetal bovine serum (Hyclone) or 10% pooled human serum (in house preparation). Of note, for functional assays and expansion, no CD3 antibody was used for cell sorting. Sorted T cells were stimulated polyclonally with 1 μg/ml phytohemagglutinin (PHA) (Sigma) in the presence of irradiated (5000 rad) allogeneic feeder cells (PBMC; 2 × 10⁵ cells per well) and 10 ng/ml of interleukin (IL)-15 (Biolegend) in a 96-well round-bottom plate (Corning/Costar). T-cell lines were maintained in complete medium with IL-15 until analysis.

**DNA preparation and TCRβ sequencing**. Sequencing of the variable V–J or V–D–J regions of TCRβ genes was performed on genomic DNA of sorted T-cell populations. DNA was extracted from circulating and tumor-resident CD8+ and CD4+ T-cell subsets ex vivo, and expanded T cells (for tumor reactivity studies), at numbers ranging from 1 × 10⁴–1 × 10⁵ cells (DNeasy Blood and Tissue Kit, Qiagen). The TCRβ CDR3 regions were sequenced and mapped using the human hsTCRB sequencing kit (ImmunoSEQ, Adaptive Biotech). Samples were sequenced using a MiSEQ sequencer (Illumina). Coverage per sample was >10×. Only data from productive rearrangements were extracted from the ImmunoSEQ Analyzer platform for further analysis. Clonality of the different T-cell subsets was assessed by nucleotide sequence comparison of all clones in each subset. To compare the TCR Vβ overlap (or similarity) of two given populations, we used Morisita's overlap index. For the comparison of shared clones between subsets, we analyzed the top 30 clones in each subset.

**Whole-exome sequencing (WES)**. For each FFPE tumor specimen, eleven 5-μm slides were cut from the FFPE block for histopathological and genome sequencing. One slide was stained with hematoxylin and eosin and was reviewed by a board-certified pathologist to identify regions of high tumor purity. These regions were then macrodissected from the corresponding ten unstained and deparaffinized slides and pooled for DNA and RNA coextraction. DNA and RNA purification was performed on an automated QiaCube instrument using DNA/RNA AllPrep reagents (Qiagen) according to the manufacturer's instructions. Corresponding normal DNA for germline exome testing was purified from peripheral blood mononuclear cells (PBMC) as above. DNA and RNA were quantified using a Qubit fluorometer (Thermo Fisher).

WES for tumor and germline specimens was performed on purified DNA as follows: DNA was prepared into tagged sequencing libraries using Kapa HyperPlus library preparation reagents (Roche) and exome hybrid-capture was performed using the xGen Research Panel kit (IDT). Captured library pools were normalized and loaded onto a HiSeq 4000 sequencer (Illumina) for next-generation sequencing. WES reads were aligned to Genome Reference Consortium Human Build 37 (hg19) followed by GATK preprocessing. Somatic mutation calling was performed using Mutect, Somaticsniper, Strelka, and Varscan. Initial filtering criteria consist of >10 tumor exome reads, >10 germline exome reads ≥10% variant-allele frequency (VAF) in the tumor exome, >10 normal reads, tumor/normal variant frequency ≥5. Somatic mutations that passed the filters were further annotated with 1000 genomes project, Exome Aggregation Consortium (ExAC), The Catalogue of Somatic Mutations In Cancer (COSMIC) databases using Annovar and SNPeff was used to predict the variant functional effect. Every mutation found to have a protein-coding change was used to build putative neoepitopes of 25-mer amino acid sequence.

Paired tumor/germline whole-exome aligned BAM files along with five other tumor and germline samples from different cases were loaded into Integrative Genomics Viewer (Broad Institute) to perform validation of identified nonsynonymous mutations and to filter potential sequencing artifacts. Mutations that are identified by ≥2 (of 4) mutation callers were manually interrogated. Potential 25-mer neoantigen sequences that passed the above filters were converted into FASTA format and run through NetMHCpan 4.0 Server (Technical University of Denmark). NetMHCpan generated 8–11mer peptides from the 25-mer neoantigen sequence and predicted binding affinity to patient-specific MHC Class-I molecules. Peptides predicted to have a binding affinity ≤0.5 nM were considered candidates for further evaluation. Those peptides were synthesized by GenScript USA Inc. and subsequently tested for reactivity against patient CD8+ T-cell subsets. The predicted mutated antigens used for reactivity screening are available in Supplementary Data 2. WES data are not publicly available due to HIPAA protection of the patients' germline sequencing data.

**RNA sequencing**. RNA purified from FFPE tissues was prepared into RNA-seq libraries using RNA Access Library Preparation reagents (Illumina) according to the manufacturer's instructions. RNA-seq libraries were assessed for quantity and quality by TapeStation (Agilent) and QuBit (Thermo Fisher). Libraries were pooled and sequenced at a depth of 25–50 million reads on a HiSeq 4000 sequencer (Illumina). RNA alignment was performed using STAR alignment, duplicate reads were marked using Picard's Mark Duplicate tool, and fragments per kb per million mapped reads (FPKM) values were calculated using cufflinks. FPKM levels were used to assess the expression of candidate mutations that are identified using whole-exome data. In select cases, candidate neoepitopes with transcript levels exceeding 100 FPKM were prioritized.

Patients did not provide consent to release read-level RNA-sequencing data containing private/rare variants, but de-identified datasets are available upon request.

**Human papillomavirus (HPV)-specific gene production**. To screen for recognition of HPV tumor antigens, four constructs were used, encoding for the full-length amino acid sequences of HPV (type 16 and 18) E6 and E7 oncoproteins (159, 99, 159, and 106 aa, respectively). They were cloned into pcDNA3.1+CEF-MHC-1-v2 using HindIII and BamHI restriction sites and flanked with AAGCTTGCCACC (5′) and GGATCC (3′) using GenScripts service. Following linearization of the constructs, DNA was cleaned up using a DNA Cleanup & Concentrator kit (Zymo Research). Next, we used 1 μg of linearized DNA to generate in vitro–transcribed (IVT) RNA using the Mmessage Mmachine T7 Ultra kit (Life Technologies) as instructed by the manufacturer. RNA was cleaned up (Zymo Research) and concentration determined prior to use in transfections.

**Transfection of HPV RNA**. Cryopreserved autologous PBMC were resuspended in Opti-MEM (Life Technologies) at 10 × 10⁶ to 30 × 10⁶ cells/ml. In total, 2 μg of HPV RNA was added to a 2-mm gap electroporation cuvette, followed by 50 μl of PBMC. Cells were electroporated at 250 V for 5 ms for one pulse, using a BTX ECM 830 Square Wave Electroporation System (Harvard Bioscience Inc.). Electroporated PBMC were washed and resuspended in complete medium at 1.3 × 10⁶–2.5 × 10⁶ cells/ml. In total, 100 μl volumes of electroporated PBMC were co-cultured with CD8+ T-cell subsets in an ELISpot assay on the same day.

**IFN-γ enzyme-linked immunospot (ELISpot) assay and detection of activation markers by flow cytometry**. The immunogenicity of predicted neoepitopes and HPV antigens against sorted CD8+ T-cell populations (DN, SP, DP) was assessed via paired ELISpot assay and detection of activation markers (CD25, 4-1BB, PD-1, Granzyme B) by flow cytometry. The PVDF membrane on ELIIP plates (Millipore, MAIPSWU10) was activated by adding 70% ethanol, 50 µl per well, for 2 minutes. Wells were washed 5× with PBS and then coated with 50 µl of 10 µg/ml IFN-γ capture antibody (Mabtech, clone: 1-D1K) overnight. For OKT3 controls, wells were coated with a mixture of IFN-γ capture antibody (10 µg/ml) and OKT3 (1 µg/ml). Cryopreserved T-cell subsets were thawed and rested overnight at 37 °C in complete medium supplemented with IL-15 (10 ng/ml). T cells were washed, and $1 \times 10^5$ T cells were added per well to a 96-well plate. In initial screenings evaluating mutated antigen recognition, peptides were pooled (12 peptides per pool) and added to wells at a final concentration of 1.25 µg/ml. In subsequent screenings, reactive pools were deconvoluted and single peptides were added to wells at 2 µg/ml. Prior to coculture, the plates were washed 1× with PBS, followed by blocking with complete RPMI media for at least 1 h at room temperature. T-cell and peptide cocultures were carried out for 20 h. In HPV-specific T-cell recognition assays, 100 µl of electroporated PBMC were added to T cells and transferred to an ELIIP plate. Following coculture, cells were harvested and assessed via flow cytometry for upregulation of activation markers. ELIIP plates were washed 6× with PBS+ 0.05% Tween-20 (PBS-T) and then incubated for 2 h at room temperature with 100 µl per well of a 0.22 µm-filtered 1 µg/ml biotinylated anti-human IFN-γ detection antibody solution (Mabtech, clone: 7-B6-1, diluent consisted of 1× PBS supplemented with 0.5% FBS). The plate was then washed 3× with PBS-T, followed by 1 h incubation with 100 µl per well of streptavidin-ALP (Mabtech, diluted 1:3000 with above diluent). The plate was washed 5x with PBS followed by development with 100 µl per well of 0.45-µm-filtered BCIP/NBT substrate solution (KPL, Inc.). The reaction was stopped by rinsing thoroughly with cold tap water. ELISpot plates were scanned and counted using an ImmunoSpot plate reader and associated software (Cellular Technology Limited). A well was considered positive for T-cell reactivity if >100 spot-forming cells (SFCs) were present, as assessed by the SmartCount function in ImmunoSpot software.

**Pathologic analysis and multiplex IHC**. Primary HNSCC tumor and lymph-node specimens were for evaluated for tumor size and the presence or absence of lymph-node metastases and staged according to the criteria of the American Joint Committee on Cancer (AJCC, seventh & eighth edition). H&E staining was performed on all tissue slides. To prepare specimens for multiplex IHC, tissue sections were cut at 4 µm from formalin-fixed paraffin-embedded blocks. All sections were deparaffinized, subjected to heat-induced epitope retrieval, and stained with the Leica Bond RX autostainer. Two multiplex IHC panels were performed on two consecutive tissue slides using the antibodies listed in Supplementary Table 5. Antigen–antibody binding was visualized with the TSA-Opal reagents from PerkinElmer. Antigen retrieval treatment was performed between antibody detection to prevent cross-reactivity. Tissue slides were incubated with DAPI as a counterstain and coverslipped with VectaShield mounting media (Vector Labs). Control tissue samples were stained for each marker separately.

**Cell quantification in high-resolution images**. Digital images were captured with the PerkinElmer Vectra-Polaris platform following hot spot lymphocyte assessment: Areas in the tumor-stroma interface were scanned at ×20 and selected for analysis. We obtained six images of 0.36 mm$^2$ each per tissue sample for analysis. Multiplexed images were analyzed with InForm Software (PerkinElmer). The total number of cells per mm$^2$ was enumerated for all the cell phenotypes expressed in the stroma and the tumor compartment. Tissue samples stained by conventional H&E were scanned with the Leica SCN400F platform at ×20 and magnified at ×200–400 for immune infiltrate evaluation.

**Reporting summary**. Further information on research design is available in the Nature Research Reporting Summary linked to this article.

## Data availability

The authors declare that the TCR sequencing data have been deposited in the ImmuneACCESS database (Adaptive Biotechnologies) under the https://doi.org/10.21417/RD2020NC and URL clients.adaptivebiotech.com/pub/duhen-2020-nc. The study protocol referenced in this study is available as Supplementary Data 1. The predicted mutated antigens are available as Supplementary Data 2. Underlying WES sequencing data are not publicly available due to HIPAA protection of the patients' germline sequencing data. We do not have patient consent to release read-level RNA-sequencing data containing private/rare variants. Any inquiries for accessing these data (including RNA-sequencing data) should be directed to eacri. bioinformatics@providence.org and we will grant access to the de-identified datasets for research purposes. Supporting statistical documentation is summarized in Supplementary Data 3. All the other data supporting the findings of this study are available within the article and its supplementary information files and from the corresponding authors upon request. Source data are provided with this paper.

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

## Acknowledgements

The authors wish to thank Raina Tamakawa and Brenda Fisher for assistance with consenting patients for the study; Michael Beymer and Cheri Goodall for processing patient samples; Will Redmond and all employees of the Immune Monitoring Lab and Miranda Gilchrist at the Flow Cytometry Core for help with processing, analyzing, and providing patient samples; the clinical research coordinator Lessli Rushforth for her assistance in receiving patient samples and data. This work was funded in part by a research grant from MedImmune, LLC to support the OX40 agonist work. MedImmune, LLC provided the IgG1 OX40 antibody but was not involved in study design, data collection, and paper writing.

## Author contributions

A.D.W. and R.B.B. initiated the trial and supervised the study. R.D., R.B.B., and A.D.W. designed the experiments. R.D., C.B.-M., and A.K.F. performed the experiments. C.B.-M. carried out multiplex immunohistochemistry staining and analysis. A.K.F. performed the neoantigen reactivity ELISpot assay with guidance from E.T. R.D., A.K.F., and A.D.W. analyzed the data. Y.K. analyzed the immune phenotyping data from the Immune Monitoring Lab. C.B.B. assessed the H&E staining and performed WES of paired tumor and blood samples. R.B.B. wrote the clinical trial protocol as principal PI, assisted in patient recruitment and sample collection. R.S.L. and B.D.C. are co-PIs on the study protocol and helped in patient recruitment. S.-C.C. did the statistical analysis. V.R. analyzed the WES data and helped with the neoantigen prediction pipeline. B.B. provided mentorship and guidance for WES and neoantigen prediction analysis. E.T. provided reagents and protocols for the HPV and neoantigen screening. R.D., R.B.B., and A.D.W. wrote the paper; W.U. carefully reviewed the paper.

## Competing interests

A.D.W. is the founder of AgonOx, which has licensed the use of OX40 agonist patents for therapeutic use in cancer patients. The remaining authors declare no competing interests.
