## [Peer Review File · Nature Communications]

Reviewers' comments:

Reviewer #1 (Remarks to the Author); expert on neoepitopes, anti tumor T-cell response, immunotherapy clinical trials:

While the results presented in this manuscript demonstrate that there is little toxicity associated with treatment of head and squamous carcinoma patients with soluble anti-OX40 antibody in the neoadjuvant setting, there were issues related to some of the additional conclusions that need to be further addressed. It was not clear how the lymphopenia seen at the time of resection and shown in Fig. 1b was related to the cell counts shown in Fig. 1c, as it appeared that for several of the patients there was a 2-fold or more drop in the lymphocyte count while the CD4, CD8 and B cell counts demonstrated little if any change at this time point. What accounted for this drop if the T cell and B cell counts showed a minimal change? The cell counts for the patients in the D8/12/19 groups should also have been separated to determine whether the day of resection influenced the lymphopenia that was observed, as this appeared to vary significantly from patient to patient. The data also should not have been presented as fold change over baseline as this will obscure the absolute differences between values, which for some parameters such as the number of Ki67+ cells, would be expected to be very low. Statements were also made in the text that the percentage of Treg cells or the proliferation of CD4+ and CD8+ T cells but there was no indication of the statistical significance of these differences. While the Ki67+ CD8+ EMRA- and EMRA+ cell frequencies appeared to increase at D12, there was no indication of whether the total frequencies of the CD8+ EMRA- and + T cells were different at this time point. The statement made in the first paragraph on page 8 that the increased CD8+ TIL proliferation was predominantly in CD103+CD39+ CD8+ TIL, but this should be supported by data provided in the manuscript. It was also not clear when the TIL were analyzed – was the phenotypic and tSNE analysis carried out on fresh tumor digest samples or following PHA stimulation, were the T cells purified as bulk T cells, and if the analyses were carried out following stimulation, at what time point was this done? The attempt to correlate that fold change in CD8+CD39/103+/Ki67+ with clinical status was also flawed in that there did not appear to be any statistically significant difference between the 4 of 4 patients judged to be IR by the fact that the fold change in CD8+CD39/103+/Ki67+ cells was greater than 1 and the 8 of 12 patients with no significant change in this ratio who have also not recurred. The potential associated between tumor mutational burden and the percentage of double positive cells prior to treatment in the 4 IR patients is driven by the single tumor in this group with the highest mutational burden, as the frequency of DP cells in the other 3 IR patients were in the range of the immune non-responder (INR) patients. It was not clear what point was being made by the reference to HPV status, as again there did not appear to be a significant difference between the PFS of patients whose tumors were or were not HPV+. Given the relatively small number of patients in these groups, it is unlikely that anything can be deduced from this analysis. The fact that one of the many parameters evaluated, the fold change in CD8+CD39/103+/Ki67+ T cells, appeared to be associated with tumor recurrence is also suspect, as this could just have occurred by chance.

Finally, the analysis of the TCR repertoire was inconclusive as there did not appear to be any statistically significant changes following anti-OX40 treatment. In addition, the fact that reactivity was found in the CD39/103 DP TIL population from 3 patients but also in the CD39 SP population from 1 of these patients does not support the contention that tumor-reactive T cells are enriched in the DP population, as 1 of 3 is not statistically significant from 3 of 3 patients.

Reviewer #2 (Remarks to the Author); expert on neoadjuvants, tumour immune response and immunotherapy clinical trials:

This is a comprehensive and interesting study from a cancer immunology group in Portland who have been studying the role of OX-40 costimulation in anti-tumor immunity. This group developed a murine anti-human OX-40 agonistic mAb that was originally tested in a mixed group of late

stage solid tumor patients, with evidence of T cell responses and some evidence of partial clinical activity. In this report, the group studied a group of head and neck squamous cell carcinoma patients who received their OX-40 mAb in the neoadjuvant setting, prior to a definitive surgical resection. They used a constant dose of antibody (3 injections) based on the previous Phase I trial, but varied the time from the end of the antibody treatment to surgery (2 weeks, 1 week, 2-3 days). There have been recent studies in other solid tumors that strongly suggest that the timing of immunotherapy before surgical resection can enhance the anti-tumor immune response and clinical benefit. This group was interested to identify the most appropriate time prior to surgery to give the antibody based on immune responses. Interestingly, they did not spend a lot of time discussing this in the paper. This may have more to do with the diverse set of patients enrolled. Clinical responses were observed in a subset of patients, but it is difficult to understand how durable they were and how such responses might be to a comparable group of patients treated with the standard of care; there was no randomized control group. However, the group performed a series of in-depth immune monitoring studies in these patients, which is really quite interesting. In particular, they identified a population of CD103+CD39+CD8+ T cells that seem to be preferentially responsive to tumors. They also could identify how induced immune responses infiltrated the parenchyma of the tumor, as opposed to just the stroma. The types of immune monitoring (multiplex IF and FACS from baseline and on-treatment biopsies, T cell receptor sequencing) are state of the art and allowed for valuable insight. The findings point to important considerations for how to study the local anti-tumor immune response in patients. These investigators have performed a significant amount of work, have generated quite a bit of interesting data, and the work makes for a concise story. The data acquisition and reliability of the data look to be quite robust, but lacks statistical power in many places. My specific concerns are outlined below.

- It is not clear why this study was done in patients with head and neck cancer? They did not look to be included in the original basket trial.
- Also, it took a long time to enroll and treat just 17 patients. Was there a problem with the enrollment?
- Figure 1. How many patients are included in each timepoint? Statistics?
- Figure 2. Interesting results. Statistical evidence for differences?
- Figure 3. Interesting re-distribution of TIL after treatment, but are the differences statistically different? Or, just too few patients?
- Figure 4. How does the OS and PFS compare to a contemporary cohort of similar patients without neoadjuvant treatment?
- Figure 6. This story of tissue resident CD8 T cells that express CD39 is very interesting and timely. How do the authors think this resonates with the recent data on how CD39 can identify bystander T cells within tumors?

Reviewer #1 (Remarks to the Author); expert on neoepitopes, anti-tumor T-cell response, immunotherapy clinical trials:

While the results presented in this manuscript demonstrate that there is little toxicity associated with treatment of head and squamous carcinoma patients with soluble anti-OX40 antibody in the neoadjuvant setting, there were issues related to some of the additional conclusions that need to be further addressed.

We thank the reviewers for the critical reading of the manuscript and their insightful comments. Below is our point-by-point reply to the concerns that were raised. We have also been working with our biostatistician, Shu-Ching Chang, to address statistical concerns that were raised by both reviewers. We are providing a document that contains the statistical analyses for Figures 1, 2 and 4. This document is not intended to be published with the manuscript, but contains comprehensive statistical analysis as requested by the reviewers and the editor.

- It was not clear how the lymphopenia seen at the time of resection and shown in Fig. 1b was related to the cell counts shown in Fig. 1c, as it appeared that for several of the patients there was a 2-fold or more drop in the lymphocyte count while the CD4, CD8 and B cell counts demonstrated little if any change at this time point. What accounted for this drop if the T cell and B cell counts showed a minimal change?

CBC analysis was performed on whole blood in the clinic for most patients at D-1 (prior to anti-OX40 treatment), Day of Surgery (D-8 or D-12 or D-19), D-34 and D-55. Results were reported as absolute counts/L and we have summarized those counts for lymphocytes and neutrophils in Figure 1b. As opposed to absolute counts, Figures 1c-1g depict the change in the percentages of viable T- and B-cells as analyzed by flow cytometry. We have pointed out the differences between Figures 1b and Figures 1c-1g in the legend for Figure 1 to clarify this to the reading audience. Since the data in Figures 1c-1g are calculated as a percentage of viable cells, they do not reflect cell numbers in the CBC analyses. Also, in the revised Figures 1c-1g, we have replaced all graphs to indicate the **change in percentages** over time by treatment group, **instead of fold-change**. The fold-change analyses have now been removed from the manuscript. The revised Supplementary Figure 1 now shows the raw values in percent for each blood sample obtained from the patients.

It has been observed that when immunotherapy drugs such as IL-2 or anti-PD-1 are administered to patients, there appears to be a transient migration of all lymphocyte subsets from the peripheral blood into lymphoid organs, followed by a rebound of all these subsets in the blood later on. Similar to these drugs, anti-OX40 treatment (in the phase I trial) was shown to cause a transient decrease in lymphocyte counts in the blood (together with a concomitant increase in neutrophils) followed by a rebound (*Curti, B.D., et al., Cancer Res, 2013*). In Figure 1b, our objective was to determine whether a similar trend occurred in the neoadjuvant setting after surgery, which we confirmed in this study.

- The cell counts for the patients in the D8/12/19 groups should also have been separated to determine whether the day of resection influenced the lymphopenia that was observed, as this appeared to vary significantly from patient to patient.

We thank the reviewer for this comment as his/her suggestion clarifies the patient stratification within Figure 1. We applied the same color code (black=D-8, orange=D-12 and blue=D-19) to Figure 1b and represented the data as average by cohort. We found that in the 4 patients who underwent surgery at D-19 (blue circles) lymphopenia was less pronounced in the peripheral blood at D-19, probably due to a rebound in lymphocyte counts at D-19. However, there was a larger decrease in lymphocyte counts observed at both D-8 and D-12 timepoints in concordance with the transient lymphopenia observed early on after anti-OX40 treatment in the initial clinical trial (*Curti BD et al., Cancer Res, 2013*).

- The data also should not have been presented as fold change over baseline as this will obscure the absolute differences between values, which for some parameters such as the number of Ki67+ cells, would be expected to be very low.

Based on the feedback by the reviewers and our biostatistician, we have now graphed the mean of the raw values, measured in percentages. All individual raw data will be available as source document as requested by *Nature Communications* so that the reading audience will be able to access all data points from these patients. We have also included the graphs of all raw data in the Supplementary Figure 1 instead of the fold-change. We found that patient baseline samples in this study range from 0.5%-5.9% Ki-67-expressing Tconv memory CD4+ and memory CD8+ T cell subsets at baseline, and are higher in Treg CD4+ cells (4%-29%).

- Statements were also made in the text that the percentage of Treg cells increased or the proliferation of CD4+ and CD8+ T cells but there was no indication of the statistical significance of these differences.

We have worked with our statistician, Shu-Ching Chang, and we have used a paired t-test to determine statistical significance of differences between D1 and D12. In addition, a generalized estimating equation (GEE) model was utilized to compare cell values over time (D1, D8/12/19, D34 and D55), to account for correlation among repeated measures from each subject. p-values are annotated as asterisks in the revised Figure 1 (based on the paired t-test) and p-values based on the GEE model are provided in a supporting document (not to be included in the manuscript).

- While the Ki67+ CD8+ EMRA- and EMRA+ cell frequencies appeared to increase at D12, there was no indication of whether the total frequencies of the CD8+ EMRA- and + T cells were different at this time point.

We have analyzed the frequencies of both the CD8+ EMRA+ and CD8+ EMRA- subsets in all patients at each timepoint. The data was added as **Figure 1e**. As shown in this graph, there are only small (non-statistically significant) changes in the frequencies of CD8+ T_{EMRA-} and CD8+ T_{EMRA+} cells over time. Baseline frequencies varied between patients, from 14.1% to 81.6% at baseline in CD8+ T_{EMRA-} cells and 3.41% to 66.4% in CD8+ T_{EMRA+} cells, but there was little change in the percentages over time within each patient sample. It appears that monitoring Ki-67 within these subsets reflects increased activation post-anti-OX40 treatment, independent of

the baseline and post-frequencies within these two CD8+ T cell subsets. As stated above, we believe that the lymphocytes are migrating from the blood into lymphoid organs and tissues/tumor post-anti-OX40 treatment and become activated as illustrated by the TIL data presented in Figure 2e.

- The statement made in the first paragraph on page 8 that the increased CD8+ TIL proliferation was predominantly in CD103+CD39+ CD8+ TIL, but this should be supported by data provided in the manuscript.

To address the issue raised by the reviewer we have now provided new data that replaces the data in Figure 2e, which originally showed the percentage of Ki67+ cells in all CD8+ TIL. We now present the proliferation data in all three CD8+ TIL subsets; Figure 2e now shows Ki-67 expression among DN (CD103-CD39-), SP (CD103+CD39-) and DP (CD103+CD39+) CD8+ T cells in all patients, before and after anti-OX40 treatment. In our previous manuscript we had shown that the DP CD8+ TIL have a higher expression of Ki-67, compared to DN and SP TIL, in samples obtained from patients that had elective surgery (*Duhen T et al., Nat Commun, 2018*). As indicated in the Figure 2e (right side), four patients (HNOX04, HNOX18, HNOX11 and HNOX15) show an increase in proliferation in the DP TIL subsets, whereas one patient (HNOX07) also displayed an increase in Ki-67 expression in the SP TIL after anti-OX40 treatment. In contrast to the CD8+ TIL, the majority of patients showed an increase in CD4+ TIL proliferation post-anti-OX40 treatment. We speculate that the increase of Ki-67 on CD4 T cells is more pronounced due to direct engagement of OX40 on CD4+ T cells, whereas other factors may be involved with increasing CD8+ T cell proliferation. Another aspect that may play a role in detecting changes in Ki67+CD8+ TIL after anti-OX40 therapy was the time when the samples was obtained after surgery, as none of the Day 8 patients showed an increase in Ki-67 in CD8 TIL.

- It was also not clear when the TIL were analyzed – was the phenotypic and tSNE analysis carried out on fresh tumor digest samples or following PHA stimulation, were the T cells purified as bulk T cells, and if the analyses were carried out following stimulation, at what time point was this done?

This point has now been clarified in the Materials and Methods section (page 20-22) for all tumor samples that were assessed in this study. Tumors were digested directly after surgery, followed by a Ficoll gradient to enrich for mononuclear cells. PBMC were isolated from blood using Ficoll gradient, and all cells were stored in LN2 until further analysis or cell sorting. For phenotyping, cryopreserved TIL and PBMC, were stained **directly ex vivo** without expansion, using two 15-color flow cytometry panels. These data were used for the tSNE analysis in Figure 2 as well as the graphs in Figure 1. For TCR repertoire analyses (Figure 5), the subsets were sorted directly *ex vivo* and then sequenced, as described in the Materials and Methods section. For the neo-antigen reactivity experiments shown in Figure 6, TIL were sorted and expanded *in vitro* for three weeks prior to the ELISpot analyses.

- The attempt to correlate the fold change in CD8+CD39/103+/Ki67+ cells with clinical status was also flawed in that there did not appear to be any statistically significant difference between the 4 of 4 patients judged to be IR by the fact that the fold change in

CD8+CD39/103+/Ki67+ cells was greater than 1 and the 8 of 12 patients with no significant change in this ratio who have also not recurred.

As the reviewer points out the results based on a Fisher exact test show that there is no significant association of response status with recurrence status (p-value = 0.2445).

	No recur, n (%)	Recur, n (%)
No response	7 (63.6)	5 (100)
Response	4 (36.4)	0 (0)

As of February 2020, five of the 12 patients that were immunologic non-responders have recurred (HNOX06, HNOX09, HNOX10, HNOX17 and HNOX19), while none of the 4 immunological responders (HNOX04, HNOX07, HNOX11 and HNOX18) have recurred. We agree that the number of patients is too small to demonstrate that CD8+CD39+CD103+Ki-67+ could be a predictive biomarker of anti-OX40's activity. However, we believe the results are hypothesis-generating and we intend to use these results to inform the design of a larger anti-OX40 study that would be adequately powered to answer this question.

- The potential association between tumor mutational burden (TMB) and the percentage of double positive cells prior to treatment in the 4 IR patients is driven by the single tumor in this group with the highest mutational burden, as the frequency of DP cells in the other 3 IR patients were in the range of the immune non-responder (INR) patients.

After reviewing the data with our biostatistician, we agree with the reviewer's comment above and have removed this data from the manuscript. While TMB has been shown to predict survival after immunotherapy across multiple cancer types in a large cohort of patients (Samstein RM et al., *Nat Genetics*, 2019), these authors acknowledged that TMB values and cut-offs vary widely across different cancer types. We conclude that with the small number of patients presented in this study we cannot support our claim and the figure has been removed.

- It was not clear what point was being made by the reference to HPV status, as again there did not appear to be a significant difference between the PFS of patients whose tumors were or were not HPV+. Given the relatively small number of patients in these groups, it is unlikely that anything can be deduced from this analysis.

We agree with the reviewer that the numbers are too small and have removed the survival analysis where patients are separated by HPV status (previously Figure 4c). However, we would like to point out that, at time of resubmission of the manuscript, only 4 out of 11 HPV- patients have recurred (36%), which is lower than the expected recurrence rate in this population of locoregionally advanced patients (50-65%) (Fakhry C et al., *J Clin Oncol*, 2014). While the number of patients is inadequate to draw any major conclusions, we believe this relatively low recurrence rate and high overall survival rate (only one death in this population) are hypothesis-generating and we have modified the manuscript accordingly (page10/11).

- The fact that one of the many parameters evaluated, the fold change in CD8+CD39/103+/Ki67+ T cells, appeared to be associated with tumor recurrence is also suspect, as this could just have occurred by chance.

As previously discussed, this small study allows for the opportunity to obtain hypothesis-generating data. The hypothesis suggested by Figures 2 and 3, is that anti-OX40 increases the CD8+CD39+CD103+Ki-67+ TIL subset either directly, by interacting with this cell population leading to increased proliferation, or indirectly, through increased CD4+ T cell help. We have focused our analysis on the CD103+CD39+Ki-67+ subset of CD8+ TIL based on our previous publication (*Duhen T et al., Nat Commun, 2018*), where we found that this subset is highly enriched for tumor-reactive T cells. The initial goal of this clinical trial was to determine safety of the neoadjuvant treatment in HNSCC as well as to explore the effects of anti-OX40 treatment on immune cell subsets in tumor biopsies pre- and post anti-OX40 treatment. We also analyzed other activation markers, and have observed increases in CD39 and ICOS (in CD4+ TIL) and increased CD38 expression (in CD8+ TIL) in the same patients. There have been several immunotherapy neoadjuvant trials (NSCLC, Melanoma) published recently, studying pre- and post-drug biopsy samples similar to our approach, but none of them have addressed whether there were increases in tumor antigen-specific T cells (*Forde PM et al., N Engl J Med, 2018; Zhang J et al., Clin Cancer Res, 2019; Amaria RN et al., Nat Med, 2018, Blank CU et al., Nat Med, 2018, Huang AC et al., Nat Med, 2019, Rozeman EA et al., Lancet Oncol, 2019*). While we agree that the numbers are small in this trial and any observation, even one associated with a $p = 0.05$ has a 1 in 20 probability of having occurred by chance, it is compelling that the four patients showing an increase in DP CD8+ TIL, have not recurred after a single cycle of anti-OX40. Furthermore, three of these four patients were HPV negative, which have a worse prognosis compared to HPV+ patients. While the data is not statistically significant it is hypothesis-generating and will be the focus of future anti-OX40 clinical trials. Ultimately, we hope this work will lead to other investigators exploring the DP CD8 TIL sub-population in future immunotherapy trials.

- Finally, the analysis of the TCR repertoire was inconclusive as there did not appear to be any statistically significant changes following anti-OX40 treatment.

TCR sequencing was performed on sorted CD4+ and CD8+ T cell subsets from paired tumor and blood samples, which is in contrast to other neoadjuvant studies, where total TIL were analyzed. While the analysis of the repertoire of all T cells pre and post would have been one plausible approach, we decided to focus on comparing the repertoire of the tumor-reactive and non-tumor reactive cells. We agree with the reviewer that we cannot demonstrate statistical differences in the TCR repertoires of responders vs non-responders because of the limited availability of pre-biopsy tissue, which in many patients was depleted after performing the phenotyping analysis. However, we would like to point out that the TCR sequencing data confirms the following:

- Clonality is increased in CD4+ T cells after anti-OX40 in the 4 patients analyzed, correlating with increases in Ki67 and suggesting expansion of specific clones in situ in the TIL.
- Clonality is increased in CD8+ T cells in the blood (3/4 patients) – which might also be reflective of the increase in proliferation that is shown in Figure 1d.

We did not detect an increase in CD4 clonality in the blood even though there was an increase in CD8 clonality in 3/4 patients. However, the clonality of CD4+ T cells is approximately one-half log to one log lower than that of CD8+ T cells, and thus any expanding clone could easily be diluted due to the polyclonal nature of the CD4+ T cells in blood.

We have removed the overlap analysis of peripheral CD4 memory cells with TIL and drLN (Figure 5d) due to the polyclonal repertoire of CD4 T cells (and conclusive overlap analysis would require much deeper sampling in the blood).

We believe it is important to show in Figure 5 d (Figure 5e previously) that CD8+ DP TIL have a unique clonal landscape and do not share many sequences with drLN, DN and SP cells (neither do they overlap with memory CD8+ T cells from the peripheral blood).

Lastly, we focused on the top clones in DP TIL as we hypothesized that they contained a high percentage of tumor-reactive cells and might be expanded post anti-OX40 treatment. However, while we observed expansion of some clones in DN and SP TIL (post vs pre) and could detect most of those clones in peripheral blood, we did not detect most of the DP TIL clones in the periphery most likely due to the depth of sequencing. Comparing pre and post DP CD8 TIL shows for both patients HNOX04 and HNOX18 that some clones decrease while others expand (which could also explain why clonality did not change in those two patients). We have re-worded the manuscript to clarify that the TCR sequencing data will be confirmed in future studies focusing on samples before and after immunotherapy treatment.

- In addition, the fact that reactivity was found in the CD39/103 DP TIL population from 3 patients but also in the CD39 SP population from 1 of these patients does not support the contention that tumor-reactive T cells are enriched in the DP population, as 1 of 3 is not statistically significant from 3 of 3 patients.

Our statement that the tumor-reactive T cells are enriched within the 103+CD39+ CD8+ T cell fraction is based on our earlier work published in *Nature Communications* (Duhon T et al., *Nat Commun*, 2018), where we found 6/6 patients showed markedly increased tumor-reactivity within this subset. While we agree with the reviewer that three patients is a relatively small number, we did find neo- and or tumor-associated antigen reactivity in all three of these patients in the DP CD8 TIL subset. One of these three patients (HNOX07) also showed neoantigen reactivity in the SP TIL. Interestingly, HNOX07 showed the greatest increase in Ki-67 expression in the SP CD8 TIL after anti-OX40 treatment (see Figure 2e) and we hypothesize that the CD103 SP tumor-reactive CD8 TIL had not yet upregulated CD39 expression at the time of surgery. In our initial publication describing the CD8+ DP TIL population we showed that in six patients (HNSCC and Melanoma) the majority of tumor-reactivity (to autologous tumor cell lines) was confined within the DP TIL population (Duhon T et al., *Nat Commun*, 2018). However, there was also a small percentage of tumor-reactivity also detected in the SP CD8 TIL population in a few of these patients, hence we also might expect some tumor-reactivity within the SP CD8 TIL.

Since our initial publication, we have collaborated with a team in the Netherlands where we found enrichment of neoantigen reactivity exclusively in the DP TIL in patients with colon cancer (van den Bulk J et al., *Genome Medicine*, 2019) (and this reference has been added to

the manuscript). In collaboration with Eric Tran, Ph.D, at our institute we have also found that the tumor-specific T cells are enriched within the CD8+ DP TIL in patients with cholangiocarcinoma, pancreatic cancer and HPV+ HNSCC patients (unpublished data). We would be happy to provide the unpublished results to the reviewer upon further request. Hence, in all the studies we have undertaken with human tumor samples we consistently find that the tumor-reactive T cells are highly enriched within the CD8+ DP TIL.

Reviewer #2 (Remarks to the Author); expert on neoadjuvants, tumour immune response and immunotherapy clinical trials:

This is a comprehensive and interesting study from a cancer immunology group in Portland who have been studying the role of OX-40 costimulation in anti-tumor immunity. This group developed a murine anti-human OX-40 agonistic mAb that was originally tested in a mixed group of late stage solid tumor patients, with evidence of T cell responses and some evidence of partial clinical activity. In this report, the group studied a group of head and neck squamous cell carcinoma patients who received their OX-40 mAb in the neoadjuvant setting, prior to a definitive surgical resection. They used a constant dose of antibody (3 injections) based on the previous Phase I trial, but varied the time from the end of the antibody treatment to surgery (2 weeks, 1 week, 2-3 days). There have been recent studies in other solid tumors that strongly suggest that the timing of immunotherapy before surgical resection can enhance the anti-tumor immune response and clinical benefit. This group was interested to identify the most appropriate time prior to surgery to give the antibody based on immune responses. Interestingly, they did not spend a lot of time discussing this in the paper.

In regards to the timing of surgery after anti-OX40 administration, we have added a short paragraph in the discussion in response to the reviewer's concern (page15). After the initial enrollment of 10 patients (HNOX01 – HNOX11, HNOX02 was not eligible), we determined that Day 12 +/- 3 days was the optimal timepoint to detect immune activation of T cells both in the blood and tumor. Once we determined this was the optimal timepoint, enrollment was continued with subsequent patients undergoing surgery approximately two weeks after anti-OX40 treatment.

This may have more to do with the diverse set of patients enrolled. Clinical responses were observed in a subset of patients, but it is difficult to understand how durable they were and how such responses might be to a comparable group of patients treated with the standard of care; there was no randomized control group.

In this study, there were no objective clinical or radiographic responses, nor was this a trial endpoint. The aim of the study was to determine the timing and characteristics of the immunological response after anti-OX40 treatment in the neoadjuvant setting. Secondary endpoints were focused on long-term outcomes but this was based on historical control groups. Thus, we did not include a randomized control group of patients receiving standard of care for this study. However, we did find that patients with robust CD8 TIL activation (N=4) remained tumor-free (see Figure 4).

However, the group performed a series of in-depth immune monitoring studies in these patients, which is really quite interesting. In particular, they identified a population of CD103+CD39+CD8+ T cells that seem to be preferentially responsive to tumors. They also could identify how induced immune responses infiltrated the parenchyma of the tumor, as opposed to just the stroma. The types of immune monitoring (multiplex IF and FACS from baseline and on-treatment biopsies, T cell receptor sequencing) are state of the art and allowed for valuable

insight. The findings point to important considerations for how to study the local anti-tumor immune response in patients. These investigators have performed a significant amount of work, have generated quite a bit of interesting data, and the work makes for a concise story. The data acquisition and reliability of the data look to be quite robust, but lacks statistical power in many places. My specific concerns are outlined below.

We thank the reviewer for summarizing our work and appreciate their comments and concerns. Below is our point-by-point response to the concerns that were raised.

As mentioned in the response to reviewer #1, we have worked with the biostatistician at our institute, Shu-Ching Chang, to also address some statistical issues raised by reviewer #2.

- It is not clear why this study was done in patients with head and neck cancer? They did not look to be included in the original basket trial.

Based on research performed at our institute (*Montler R et al., Clin Transl Immunol, 2016; Bell RB et al., Oral Oncol, 2016*), we found high and reproducible expression of OX40 on TIL in head and neck cancer surgical specimens, with CD4+ T cells expressing the highest levels. Expression was high on regulatory T cells and a bit lower on CD4+ TIL. A study from our group has shown that in preclinical models, treatment with anti-OX40 increases the effector function of CD4+ T conventional cells, as well as effector cytokine production (IFN- γ , TNF- α) by Treg cells (*Polesso F et al., J Immunol, 2019*). Hence, because of the high and reproducible levels of OX40 expression in HNSCC cancer patients we designed the neoadjuvant study for this patient population, as we hypothesized we might see the most reproducible immune stimulatory effects. It should be noted that we have a second ongoing trial with a humanized anti-OX40 antibody in the neoadjuvant setting enrolling patients with both melanoma and H&N cancer (NCT03336606).

- Also, it took a long time to enroll and treat just 17 patients. Was there a problem with the enrollment?

Patient enrollment in this trial was delayed due to 2 factors:

- After enrolling the first patient, the antibody supply did not pass its release criteria, thus a new lot of anti-OX40 (same clone) was manufactured for the trial (by MedImmune/AstraZeneca). It took 13 months to get the new antibody supply, thus no patients were enrolled during this interval. In reality, the first 10 patients were enrolled in a six-month period. The details for all patients are included in the source document as requested by *Nature Communications*.
- A second delay in accrual was introduced when we assessed immune parameters after the first 10 patients (HNOX01-HNOX11) to inform optimal timing in the expansion cohort, which was chosen to be Day12 +/- 3 days. The protocol was amended accordingly, which took several months. Enrollment then continued 6 months later with HNOX13. The last 7 patients completed accrual within a six-month timeframe.

- Figure 1. How many patients are included in each timepoint? Statistics?

We have included the information for all patients within an additional supplementary table (**Supplementary Table 1a**). For immune phenotyping, all 16 patients (D8, D12 and D19 cohorts)

are included in Figure 1, omitting the first patient (HNOX01, D26). As also requested by reviewer 1, we have modified Figure 1 and are now displaying the mean of the raw data instead of fold change. As mentioned to reviewer 1, we have used a paired t-test to determine statistical significance of differences between D1 and D12. In addition, a generalized estimating equation (GEE) model was utilized to compare cell values over time (D1, D8/12/19, D34 and D55), to account for correlation among repeated measures from the subjects. The raw percentages for each patient can be found in the source document for Nature Communications and are graphed in a revised Supplementary Figure 1 for each patient.

- Figure 2. Interesting results. Statistical evidence for differences?

We have performed statistical analyses on Figures 2a, d and e and have modified the figure to display raw values only. A paired t-test analysis for Figure 2a shows that the changes in CD4+ Tconv, CD4+ Treg and CD8+ T cells are not statistically significant. The same is true for Figure 2d, as some patients exhibited an increase in these markers while others show a decrease. In Figure 2e, Ki-67 expression is significantly increased in DP TIL compared to SP TIL, with an increase in proliferation pre vs post in only 4 patients. We did find that proliferation was increased significantly in CD4+ TIL (Figure 2e). We acknowledge that the number of patients is too small to detect statistically significant changes in CD8 T cells, and we believe that the observations are hypothesis generating and will inform ongoing and future clinical trials.

- Figure 3. Interesting re-distribution of TIL after treatment, but are the differences statistically different? Or, just too few patients?

Unfortunately, with only four patients that were deemed immunologic responders, we do not have statistical significance for the redistribution of proliferating CD8+CD103+ TIL after treatment. We are currently enrolling patients in a neoadjuvant study with a humanized anti-OX40 antibody (NCT03336606) and plan to enroll a larger cohort (up to 35 patients) and extend the same analyses on those tumors.

- Figure 4. How does the OS and PFS compare to a contemporary cohort of similar patients without neoadjuvant treatment?

There were 11 HPV-negative patients enrolled in this study, which had locally advanced disease (stage III or IVA by AJCC 8th Ed) and thus, we would expect a 50-55% 3-year OS and DFS rate following standard of care treatment. At the time of revision, 8 of 11 HPV- patients are alive (73%), which is higher than the expected rate in this population.

For the 6 HPV-positive patients in this study, historic data show a 3-year OS and DFS of >90% with standard of care therapy. At this time, OS is 100% in these 6 patients.

- Figure 6. This story of tissue resident CD8 T cells that express CD39 is very interesting and timely. How do the authors think this resonates with the recent data on how CD39 can identify bystander T cells within tumors?

Based on the data from our recent manuscript (*Duhen et al., Nat Commun, 2018*), we believe that CD103 expression (Integrin αE , a molecule expressed on resident memory T cells) enriches for T cells that have received signals through TGF- β in the tumor microenvironment.

Additionally, we showed that chronic stimulation through the TCR leads to upregulation of CD39. Therefore, we feel the TIL in the tumor parenchyma are recognizing neo- or tumor-associated (e.g. HPV) antigens and consequently upregulating CD39 expression. Thus, as proposed by both ourselves and Evan Newell's group in Nature (*Simoni Y et al., Nature, 2019*) **CD39 appears to be a marker that can be used to enrich for tumor-reactive TIL especially in conjunction with CD103.** However, lack of CD39 identifies bystander/cancer-unrelated CD8 TIL (Figure 3; *Simoni Y et al, Nature, 2019*).

The data in Figures 3a & 6 infer that after anti-OX40 treatment there is an expansion of the tumor-reactive subset of CD8 TIL as shown by an increased percentage of CD103+CD39+ DP T cells.

REVIEWER COMMENTS

Reviewer #1 (Remarks to the Author):

The authors adequately addressed several of the points raised by the reviewers, and the additional statistical analysis was very helpful in evaluating the results; however, there are several remaining issues regarding the original data provided in the manuscript, as well as several issues raised by the new data that was provided. First, the text on page 6 indicates that there was an increase in the proliferation of CD4+ Tconv cells between D12 and D34, but Fig. 1d and Sup. Fig. 1c,f appear to indicate that there was an increase in the proliferation between baseline and the day of surgery in the group that received surgery on day 12. It was also not apparent from Fig. 1d only the day 12 group was significant (orange asterisks) and not the day 19 group, as these curves tracked each other very closely in both the CD4+ Tconv and CD8 populations. The data shown in Sup. Fig. 1d,e also appear to indicate that the frequencies of OX40+ T cells are present on PBMC samples from some patients are greater than 50%, which is not consistent with the authors' and other investigators' studies indicating that these frequencies are generally less than 10% of peripheral T cells. The ordinate in Sup. Fig. 1e also ranges between 0 and 150, which is not appropriate for labeling frequencies. Other than the fact that it may have been more convenient to obtain peripheral blood at the time of surgery, it is not clear why the study was set up in this way, as it would have been much more useful to obtain peripheral blood at multiple time points prior to surgery in the day 12 and day 19 groups, which would have allowed conclusions about the peak of activation in individual patients rather than comparing different patients at these early time points. A major issue that remains, however, is the use of what appears to be a post hoc analysis to identify changes in T cell populations that are associated with the treatments and between T cell parameters and tumor progression. The changes in the frequencies of CD8+CD39+CD103+ and CD4+CD39+ following anti-OX-40 treatment may not have resulted from the treatment, as they appear to be random (Fig. 2 d). The generation of an IR and INR parameter that is based upon one set of parameters, fold change (pre/post) CD8+/CD39+/CD103+/Ki-67+ in tumor samples that is then associated with disease free survival is also a post hoc argument. This is not statistically valid, as multiple T cell parameters could be evaluated until 1 was found that was associated with tumor progression, but in this case Bonferroni's correction would need to be carried out to determine the significance of this finding. In addition, the p value noted in Fig. 4b was 0.2, which is not statistically significant.

Reviewer #2 (Remarks to the Author):

This is a revised submission from a group studying the results of a Phase 1b trial using an agonistic antibody to OX-40 in a group of head and neck squamous cell carcinoma patients. The original reviews were positive, but both reviewers commented on the statistical rigor, or the lack thereof, for much of the analyses. In their response, the authors were quite responsive to the critiques and actually brought a statistician onto the paper with new analyses. Some of the figures remained and some were withdrawn. In the end, they did find some statistically significant findings, and this will help the validity of the results. Other endpoints were not significant, but the authors have argued to keep the data there because of the trends and significance. In the end, the trial was small enough and did not see enough true immunologic responders to make some solid conclusions. However, the findings are still important, and will likely serve as a foundation for their future Phase II trials with a larger enrollment.

Reviewer #4 (Remarks to the Author):

Most of the analyses appear to be appropriate but some changes/clarifications would improve the presentation

- a) in the GEE analysis I assume that the correlation structure is exchangeable (compound symmetric) - this should be stated
- b) The median follow-up of 34 months: How I this calculated. I presume this was obtained using the reverse Kaplan-Meier method & if so I don't understand what the +/- 11.3 months refers to. If not then the this is not the median follow-up (p10)
- c) Not sure that the crude PFS and OS proportions (82% and 71%) have any meaning in the presence of censored observations - these proportions assume *all* patients have been all been followed-up for the same time period. They are misleading. A more meaningful measure would be the hazard (%risk of failure) obtained as the ratio of (the number of events)/(total follow-up time).

REVIEWER COMMENTS – 2nd submission

Reviewer #1 (Remarks to the Author):

The authors adequately addressed several of the points raised by the reviewers, and the additional statistical analysis was very helpful in evaluating the results; however, there are several remaining issues regarding the original data provided in the manuscript, as well as several issues raised by the new data that was provided.

We thank the reviewer for carefully reading the revised manuscript and the changes we made based on his/her comments. Below, we will address the questions raised by the new data as well as address points regarding the original submission.

First, the **text on page 6** indicates that there was an **increase in the proliferation of CD4+ Tconv cells between D12 and D34**, but **Fig. 1d and Sup. Fig. 1c,f** appear to indicate that there was an **increase in the proliferation between baseline and the day of surgery** in the group that received surgery on **D12**.

We agree with the reviewer that this is not what the figures show and have modified the sentence to “we detected an increase in proliferation between baseline and D12 and D19 in CD4+ Tconv cells.”

It was also **not apparent from Fig. 1d only the day 12 group was significant** (orange asterisks) and **not the day 19 group**, as these curves tracked each other very closely in both the CD4+ Tconv and CD8 populations.

In Figure 1d, the change in the D19 group is indeed significant. We have focused our analysis on the D12 group because of the larger number of patients (n=9). The D8 and D19 groups comprise only 3 and 4 patients respectively, thus the statistics have less power, and cannot be calculated for the D8 group. Upon the reviewer’s request we have added statistics for the D19 patients to **all** graphs in Figure 1 and have modified the text and figure legends accordingly.

The data shown in **Sup. Fig. 1d,e** also appear to indicate that the **frequencies of OX40+ T cells are present on PBMC samples** from some patients are greater than 50%, which is **not consistent with the authors’ and other investigators’ studies** indicating that these frequencies are generally less than 10% of peripheral T cells.

We agree with the reviewer’s comment and have re-analyzed the OX40 expression data. The data that was initially provided for OX40 expression was gated on **memory CD4+ T conv cells**, and the percent OX40 positive cells was based on FMO (fluorescence minus one) controls. Those parameters are not always perfect, especially in a multicolor panel. We have re-analyzed the data set for all patients, using **naïve CD4+ T conv cells** to inform the gating for negative vs positive populations, as naïve cells express very little OX40 (*Croft et al., Immunol Rev, 2009*). The data in Supp. Fig. 1d,e now represents the % of OX40+ cells among **all** CD4+ T conv cells (not just memory CD4+ T cells), which is in line with what has been previously reported. Furthermore, OX40 assessment in this manuscript used fresh whole blood, whereas our previous manuscripts used Ficoll-enriched PBMC for phenotyping, which may also explain differences between the two studies.

Representative histograms of the OX40 staining on 4 patients are included below, to visualize the difference of OX40 expression between memory and naïve CD4+ T cells.

The ordinate in **Sup. Fig. 1e** also ranges between 0 and 150, which is **not appropriate for labeling frequencies**.

We thank the reviewer for pointing out this error, we have now corrected this. As mentioned in the previous comment, we have reanalyzed the data in Supplementary Figure 1, which now appropriately reflects the levels of OX40 expression among total CD4+ Tconv cells, ranging between 4-30%.

Other than the fact that it may have been more convenient to obtain peripheral blood at the time of surgery, **it is not clear why the study was set up in this way**, as it would have been much **more useful to obtain peripheral blood at multiple time points prior to surgery** in the day 12 and day 19 groups, which would have allowed conclusions about the peak of activation in individual patients rather than comparing different patients at these early time points.

The study was set up to determine the optimal interval for immune activation between anti-OX40 administration and the time of surgery – comparing baseline blood samples with day of surgery (and later follow-up appointments) and matched tissue samples to examine the immune activation within the tumor (comparing baseline and day of surgery). We agree that it would have been informative to also obtain blood from D8 patients at a later timepoint (between D12 and D19) and conversely, blood from D19 patients at an earlier timepoint. However, it would have been very difficult to obtain these samples from each patient as it would have been

logistically more challenging (at least one extra visit for a blood draw only) and hence it was not written into the protocol.

A **major issue** that remains, however, is the use of what appears to be a **post hoc analysis** to identify changes in T cell populations that are associated with the treatments and between T cell parameters and tumor progression. The **changes in the frequencies of CD8+CD39+CD103+ and CD4+CD39+ following anti-OX-40 treatment may not have resulted from the treatment, as they appear to be random (Fig. 2 d)**. The **generation of an IR and INR parameter** that is **based upon** one set of parameters, **fold change (pre/post) CD8+/CD39+/CD103+/Ki-67+** in tumor samples that is then **associated with disease free survival** is also a post hoc argument. **This is not statistically valid**, as multiple T cell parameters could be evaluated until 1 was found that was associated with tumor progression, but in this case **Bonferroni's correction would need to be carried out to determine the significance of this finding**.

In addition, the **p value noted in Fig. 4b was 0.2**, which is **not statistically significant**.

We agree with the reviewer that post-hoc analyses are exploratory in nature only to be considered in the context of the known specific study population. In regard to the CD8+/CD103+/CD39+/Ki-67+ T cell population - we had previously published that these cells are highly enriched for tumor-reactivity, which we feel makes them a highly relevant population to follow after immunotherapy treatment (<https://www.nature.com/articles/s41467-018-05072-0>). In the same manuscript we also showed that increased percentages of these TIL (prior to conventional treatments) led to greater survival in H&N cancer patients. Therefore, we hypothesized that increasing this population after immunotherapy treatment might be beneficial. As the reviewer pointed out, the correlation between survival and increase in the CD8+/CD39+/CD103+/Ki-67+ TIL population was not statistically significant; however, it is intriguing that none of the patients (0/4) with an increase in this population have recurred. Review #2 wrote, "In the end, the trial was small enough and did not see enough true immunologic responders to make some solid conclusions. However, the findings are still important, and will likely serve as a foundation for their future Phase II trials with a larger enrollment." We agree with reviewer #2's assessment, as these results were meant to be hypothesis-generating and exploratory in nature and our hope would be that this post-hoc analysis may encourage others to test similar hypotheses with other immune stimulatory drugs (e.g. anti-PD-1).

It should be noted that in Fig 2d it is interesting that some patients showed an increase in these TIL populations while others showed a decrease in these populations, but in the end, there was no statistical significance. This data is shown to highlight the patient to patient variation that we observed and we agree with the reviewer that these trends may not have resulted from anti-OX40 treatment as we had too few patients in this study to understand the statistical significance.

Reviewer #2 (Remarks to the Author):

This is a revised submission from a group studying the results of a Phase 1b trial using an agonistic antibody to OX-40 in a group of head and neck squamous cell carcinoma patients. The original reviews were positive, but **both reviewers commented on the statistical rigor**, or the lack thereof, for much of the analyses. In their response, the authors were quite responsive to the critiques and actually brought a statistician onto the paper with new analyses. Some of the figures remained and some were withdrawn. In the end, they did find some statistically significant findings, and this will help the validity of the

results. Other endpoints were not significant, but the authors have argued to keep the data there because of the trends and significance. In the end, the **trial was small enough and did not see enough true immunologic responders to make some solid conclusions**. However, the findings are still important, and will likely serve as a foundation for their future Phase II trials with a larger enrollment.

We thank the reviewer and agree that, albeit a small study, we believe that the findings are of value, especially in light of follow up studies and future phase II clinical trials.

Reviewer #4 (Remarks to the Author):

Most of the analyses appear to be appropriate but **some changes/clarifications would improve the presentation**

a) in the GEE analysis I assume that the **correlation structure is exchangeable (compound symmetric) - this should be stated**

To clarify this assumption, we have included the description of the GEE model (below) into the manuscript.

“A generalized estimating equation (GEE) model was utilized to compare measurements over time in order to account for correlation among repeated measures from the same subject. Correlation structures including exchangeable, autoregressive with order 1 or AR (1), and independence were evaluated using quasi-likelihood under the independence model for criterion (QIC) for analysis. AR (1) was finally selected for the GEE model since it had the smallest QIC value.”

b) The **median follow-up of 34 months: How is this calculated**. I presume this was obtained using the reverse Kaplan-Meier method & if so I **don't understand what the +/- 11.3 months refers to**. If not then the this is not the median follow-up (**p10**).

We agree with the reviewer and have removed the median follow-up.

c) **Not sure that the crude PFS and OS proportions (82% and 71%) have any meaning in the presence of censored observations** - these proportions assume *all* patients have been all been followed-up for the same time period. They are **misleading**. A **more meaningful measure** would be the **hazard (%risk of failure) obtained as the ratio of (the number of events)/(total follow-up time)**.

We included the below description into the manuscript for better clarification, with the corresponding table, presenting the percentage at each time point of the Kaplan-Meier curves. “The overall and disease-free survival for the entire study population was 94% and 76% at 500 days and 82% and 71% at 1000 days, respectively (**Fig. 4a, Supplementary Table 4**).”

Below is the OS (%) and DFS (%) at each time point as depicted in Figure 4a, and **now included as Supplementary Table 4**.

0-day	250-day	500-day	750-day	1000-day	1250-day
-------	---------	---------	---------	----------	----------

OS (%)	100 (100, 100)	94 (84, 100)	94 (84, 100)	88 (74, 100)	82 (65, 100)	82 (65, 100)
DFS (%)	100 (100, 100)	82 (66, 100)	76 (59, 100)	71 (52, 96)	71 (52, 96)	71 (52, 96)

REVIEWER COMMENTS

Reviewer #1 (Remarks to the Author):

While the authors have addressed many of the concerns raised by the reviewer, there are still some concerns that need to be further addressed.

There are still some questions about the analysis of the frequencies of OX40+ TConv cells. While analyzing the frequencies of OX40+ TConv CD4+ T cells among total T cells rather than among the memory population reduces the frequencies of these T cells there are samples containing >20% positive cells, which also appears to be significantly higher than previously reported. The claim that this is due to analysis of fresh whole blood rather than Ficoll-enriched PBMC raises the question of which of these results more closely represent the true frequency of positive T cells. Given the importance of accurately measuring the frequency of these cells it is contingent on the authors to provide an analysis of the effects of different cell preparation protocols on this measurement, which would be very simple to carry out. A comparison should also have been carried out between the frequencies among individual patients using a paired t-test rather than the bulk analysis provided in Supplementary Fig. 1e where any differences between different samples groups could have been obscured.

There is also a question about the timing of the effects, which were generally seen in the day 12 and day 19 groups but not the day 8 group. It seems likely that these effects would have been seen in the day 8 group if samples had been collected 12 or 19 days after antibody administration, and that the effects were not influenced by the timing of antibody administration relative to tumor resection. Thus, the kinetics of the changes in cell populations could have been identical in all the groups. I understand that it is not possible to obtain these samples from this group of patients, but future studies should attempt to address this issue, and a statement should be added to the text indicating the possibility that these differences were a result of differences in the timing of sample collections.

There is also the bigger question of why these effects took so long to manifest in the patients. It would be helpful to know if there are any additional studies, even in murine tumor models, that have demonstrated similar kinetics.

There is also an error in the statistical analysis provided on line 152 of page 7 where a p value of 2289 was reported.

Reviewer #4 (Remarks to the Author):

I thank the authors for their response but I still have some outstanding concerns

a) There is no scientific rationale provided for the choice of the AR(1) correlation structure - why should the correlation decrease by a k-fold factor between observations on a log scale particularly as the time between observations is not uniform over all patients. At least a compound symmetric correlation provides generalizability and a more sensible interpretation. The authors should provide a-priori evidence of why the AR(1) structure is relevant. Otherwise, rework the analysis using exchangeable correlation

b) My suggestion was not to omit the median follow-up - it was to calculate it appropriately. The median follow-up is a measure of data maturity and should be provided

c) In line with (b) the completeness of follow-up (Altman et al 2002, Lancet) should also be provided. While the DFS/OS proportions at key time points are helpful, they still don't reveal the maturity of the data being analysed. Alternatively, the authors could present the data according to the guidelines given in GebSKI et. al., Int Jour Epid 2018.

d) Could the time points be in months rather than days. It is difficult for readers to immediately that 1250 days is 3.5 yrs

Reviewer #1 (Remarks to the Author):

While the authors have addressed many of the concerns raised by the reviewer, there are still some concerns that need to be further addressed.

There are still some questions about the analysis of the frequencies of OX40+ TConv cells. While analyzing the frequencies of OX40+ TConv CD4+ T cells among total T cells rather among the memory population reduces the frequencies of these T cells there are samples containing >20% positive cells, which also appears to be significantly higher than previously reported. The claim that this is due to analysis of fresh whole blood rather than Ficoll-enriched PBMC raises the question of which of these results more closely represent the true frequency of positive T cells. Given the importance of accurately measuring the frequency of these cells it is contingent on the authors to provide an analysis of the effects of different cell preparation protocols on this measurement, which would be very simple to carry out.

We have analyzed OX40 expression on cryo-preserved PBMCs in the past and, as the reviewer has pointed out, observed low expression of OX40 on naïve and resting memory CD4+ T cells. Our Immune Monitoring Core, which runs several different panels to comprehensively analyze activation markers, transcription factors, chemokine receptors and other proteins, has established a staining on whole blood without the need for Ficoll separation, freezing, or other cell manipulations. We believe that for surface markers such as chemokine and cytokine receptors as well as certain activation markers, the immediate, same day staining more closely reflects the “real” expression in the blood of these patients.

Upon the reviewer’s suggestion, we have reanalyzed several patients that were part of this neo-adjuvant trial, and re-examined OX40 expression on cryo-preserved PBMCs, using two different anti-human OX40 clones ACT35 and L106. In both cases expression was lower, than what was observed during straight ex vivo examination. Therefore, we believe the direct, ex vivo analysis of OX40 expression represents a more accurate representation of the data. The direct ex vivo analyses were performed under highly validated settings using fluorescence minus one controls.

A comparison should also have been carried out between the frequencies among individual patients using a paired t-test rather than the bulk analysis provided in Supplementary Fig. 1e where any differences between different samples groups could have been obscured.

We have re-analyzed the data based on the reviewer’s suggestion, and none of these comparisons are significant (see table below). The data are separated by timepoint and are included below. We have now color-coded supplementary figure 1e as well.

p values from paired t-test (parametric)

	D1 v. DOS	DOS v. D34	DOS v. D55
Day 8	0.5168	0.2756	0.4948
Day 12	0.4410	0.7179	0.3052
Day 19	0.3596	0.3046	0.3875

There is also a question about the timing of the effects, which were generally seen in the day 12 and day 19 groups but not the day 8 group. It seems likely that these effects would have been seen in the day 8 group if samples had been collected 12 or 19 days after antibody administration, and that the effects were not influenced by the timing of antibody administration relative to tumor resection. Thus, the kinetics of the changes in cell populations could have been identical in all the groups. I understand that it is not possible to obtain these samples from this group of patients, but future studies should attempt to address this issue, and a statement should be added to the text indicating the possibility that these differences were a result of differences in the timing of sample collections.

We agree with the reviewer that the patients that received surgery at D8 might have demonstrated a similar increase in the peripheral blood, if we would have drawn their blood at D12 and/or D19 post-anti-OX40. However, it is unreasonable to ask patients that have just undergone oral and/or maxillofacial surgery to give blood 4-10 days post-surgery. We will assure that in future trials (especially in the neoadjuvant setting) blood draws and surgery are timed more appropriately to discern and compare treatment effects. We have added a sentence to the manuscript (page 15) to point out the reviewers' concern. The manuscript already included a remark, stating that the lack of activation in the tumor microenvironment, especially in the D8 group, might be due to timing (highlighted in teal in the manuscript).

There is also the bigger question of why these effects took so long to manifest in the patients. It would be helpful to know if there are any additional studies, even in murine tumor models, that have demonstrated similar kinetics.

In the initial phase I trial with the same anti-OX40 Ab MEDI6469 (Curti et al., Cancer Res., 2013, 73(24);1-10) we performed detailed kinetic analyses of Ki-67 expression in the blood on CD4+ Tconv cells, CD4+ Tregs and CD8+ T cell subsets. Similar to the data presented in this manuscript there was a delayed kinetics, with the CD4+ Foxp3- population peaking 14 days after administration of the drug and a similar finding for CD8+ T cells (Figures 2C, D, & E, Curti et al.). In the neoadjuvant study we used the 0.4mg/kg dose, which induced the greatest increase in CD4+ and CD8+ T cell proliferation in the aforementioned phase I study and in both studies similar kinetics were observed. Since anti-OX40 agonists increase several cytokines, IL-2 in particular, the effect is most likely indirect and therefore delayed T cell proliferation was observed in both clinical studies.

There is also an error in the statistical analysis provided on line 152 of page 7 where a p value of 2289 was reported.

The error was corrected (the p value is 0.2289), see page 7. Thank you for pointing this out.

Reviewer #4 (Remarks to the Author):

I thank the authors for their response but I still have some outstanding concerns

a) There is no scientific rationale provided for the choice of the AR(1) correlation structure - why should the correlation decrease by a k-fold factor between observations on a log scale particularly as the time between observations is not uniform over all patients. At least a compound symmetric correlation provides generalizability and a more sensible interpretation. The authors should provide a-priori evidence of why the AR(1) structure is relevant. Otherwise, rework the analysis using exchangeable correlation

We have previously responded to this particular question and will reiterate why we chose to use the AR(1) structure:

The AR(1) structure was chosen because it had the **smallest QIC values** for all markers, compared to exchangeable structure. In fact, **the results and conclusions matched closely** based on either correlation structure. In the last resubmission we inserted the sentence below into the text of the Methods section – statistical analysis - to address this concern and have now added two references that explain our choice of the AR(1) structure.

“A generalized estimating equation (GEE) model [1, 2] was utilized to compare measurements over time in order to account for correlation among repeated measures from the same subject. Correlation structures including exchangeable, autoregressive with order 1 or AR (1), and independence were evaluated using quasi-likelihood under the independence model for criterion (QIC) for analysis. AR (1) was finally selected for the GEE model since it had the smallest QIC value for all markers.”

Reference:

1. Liang KY, Zeger SL. Longitudinal data analysis using generalized linear models. *Biometrika*. 1986 Apr 1;73(1):13-22.
2. Cong Xu, Zheng Li and Ming Wang (2018). *wgeesel: Weighted Generalized Estimating Equations and ModelSelection*. R package version 1.5. <https://CRAN.R-project.org/package=wgeesel>

b) My suggestion was not to omit the median follow-up - it was to calculate it appropriately. The median follow-up is a measure of data maturity and should be provided

The median follow-up was re-calculated using the reverse Kaplan-Meier method as published by Shemper and Smith (1996) [3]. The median follow-up time is **39 months with a 95% CI of 34-45 months**. We have added a sentence in the manuscript within the results section that now clarifies this issue, and have included the reference as well (page 10/11).

Reference:

3. Schemper M, Smith TL. A note on quantifying follow-up in studies of failure time. *Control Clin Trials*. 1996; **17**: 343-346

c) In line with (b) the completeness of follow-up (Altman et al 2002, Lancet) should also be provided. While the DFS/OS proportions at key time points are helpful, they still don't reveal the maturity of the data being analysed. Alternatively, the authors could present the data according to the guidelines given in GebSKI et. al., *Int Jour Epid* 2018.

Thank you for the suggestions, and we agree with the reviewer that the current presentation of the data does not reveal the maturity of the data being analyzed. Accordingly, we provide the number at risk for OS and DFS in Figure 4a, below the graph, as well as in Supplementary Table 4b. We reference Figure and Table in the results section (page 10).

Time (months)	OS				DFS			
	survival estimate	Actual # at risk	Minimum n*	Decrease in the % survival estimate for one extra event	survival estimate	Actual # at risk	Minimum n*	Decrease in the % survival estimate for one extra event
0	1	17		5.88	1	17		5.88
6	1	17		5.88	0.82	15	6	5.49
12	0.94	16	11	5.88	0.76	13	5	5.88
18	0.94	16	11	5.88	0.71	12	4	5.88
24	0.88	15	7	5.88	0.71	12	4	5.88
30	0.82	13	6	6.3	0.71	11	4	6.42
36	0.82	8	6	10.24	0.71	6	4	11.76
42	0.82	5	6	16.39	0.71	3	4	23.53
48	0.82	1	6	81.93	0.71	1	4	70.59
54	0.82	1	6	81.93	0.71	1	4	70.59
60	0.82	1	6	81.93	0.71	1	4	70.59

Supplementary Table 4b. Actual number of patients at risk based on GebSKI et al.^(*)

d) Could the time points be in months rather than days. It is difficult for readers to immediately that 1250 days is 3.5 yrs

We have reformatted Figure 4 to indicate months rather than days, as well as Supplementary Table 4a to show half-year increments for clarity.

	0-year	0.5-year	1-year	1.5- year	2- year	2.5- year	3-year
OS (%)	100 (100, 100)	100 (100, 100)	94 (84, 100)	94 (84, 100)	88 (74, 100)	82 (65, 100)	82 (65, 100)
DFS (%)	100 (100, 100)	82 (66, 100)	76 (59, 100)	71 (52, 96)	71 (52, 96)	71 (52, 96)	71 (52, 96)

Supplementary Table 4a. OS and DFS at different timepoints

REVIEWER COMMENTS

Reviewer #1 (Remarks to the Author):

I believe that the reviewers' comments have been adequately addressed in the revised manuscript.

Reviewer #4 (Remarks to the Author):

Much of my reservations have been addressed - appreciate this. However, two issues are still of concern and should be attended to:

a) If an exchangeable correlation structure in the GEE give almost the same results as the AR, then this model is the more appealing and most likely to be applicable. As there is absolutely no a-priori reason for AR(1) structure (apart from data dependent fitting) my suggestion would be to report the results from the exchangeable model - AR(1) is confusing to readers (and statisticians as well)!

b) My understanding of data maturity is 'how much information is there available at a particular time point T NOW compared to the information available if ALL patients have been followed up to T' - the number at risk at T does not address this question. This is not revealed in supplementary table 4b. Suggest that this table be modified as follows: For both OS and DFS

(i) replace values in the 'minimum n*' column with '%information available'.

(ii) remove '% decrease"

Suggest that

REVIEWER COMMENTS, 4th submission

Reviewer #4 (Remarks to the Author):

Much of my reservations have been addressed - appreciate this. However, two issues are still of concern and should be attended to:

a) If an exchangeable correlation structure in the GEE give almost the same results as the AR, then this model is the more appealing and most likely to be applicable. As there is absolutely no a-priori reason for AR(1) structure (apart from data dependent fitting) my suggestion would be to report the results from the exchangeable model - AR(1) is confusing to readers (and statisticians as well)!

We have compared the results from both correlation structures (autoregressive with order 1 and exchangeable) when submitting our response previously, and found the estimates to be the same using either model. Therefore based on the reviewer's request, we have now changed the text to indicate that we used the exchangeable model.

b) My understanding of data maturity is 'how much information is there available at a particular time point T NOW compared to the information available if ALL patients have been followed up to T' - the number at risk at T does not address this question. This is not revealed in supplementary table 4b.

Suggest that this table be modified as follows: For both OS and DFS

(i) replace values in the 'minimum n*' column with '%information available'.

(ii) remove '% decrease"

We agree and have modified the table based on the reviewer's request.

Time (months)	OS			DFS		
	survival estimate	Actual # at risk	% information available	survival estimate	Actual # at risk	% information available
0	1	17		1	17	
6	1	17		0.82	15	100
12	0.94	16	100	0.76	13	100
18	0.94	16	100	0.71	12	100
24	0.88	15	100	0.71	12	100
30	0.82	13	97.3	0.71	11	100
36	0.82	8	97.3	0.71	6	100
42	0.82	5	97.3	0.71	3	100
48	0.82	1	97.3	0.71	1	100
54	0.82	1	97.3	0.71	1	100
60	0.82	1	97.3	0.71	1	100

Actual number of patients at risk based on GebSKI et al.^(*)